# Deviations in RSV epidemiological patterns and population structures in the United States following the COVID-19 pandemic

Estefany Rios-Guzman [1,2], Lacy M. Simons [1,2], Taylor J. Dean[1,2], Francesca Agnes[1,2], Anna Pawlowski [3], Arghavan Alisoltanidehkordi[1,2], Hannah H. Nam[4], Michael G. Ison[5], Egon A. Ozer [1,2], Ramon Lorenzo-Redondo [1,2] & Judd F. Hultquist [1,2] ✉

Respiratory Syncytial Virus (RSV) is a leading cause of acute respiratory tract infection, with the greatest impact on infants, immunocompromised individuals, and older adults. RSV prevalence decreased substantially in the United States (US) following the implementation of COVID-19-related non-pharmaceutical interventions but later rebounded with abnormal seasonality. The biological and epidemiological factors underlying this altered behavior remain poorly defined. In this retrospective cohort study from 2009 to 2023 in Chicago, Illinois, US, we examined RSV epidemiology, clinical severity, and genetic diversity. We found that changes in RSV diagnostic platforms drove increased detections in outpatient settings post-2020 and that hospitalized adults infected with RSV-A were at higher risk of intensive care admission than those with RSV-B. While population structures of RSV-A remained unchanged, RSV-B exhibited a genetic shift into geographically distinct clusters. Mutations in the antigenic regions of the fusion protein suggest convergent evolution with potential implications for vaccine and therapeutic development.

The implementation of non-pharmaceutical interventions (NPIs) to limit the spread of COVID-19, such as masking, school closures, and social distancing, had dramatic impacts on the prevalence of other respiratory pathogens, including influenza viruses and respiratory syncytial virus (RSV)[1]. According to the Centers for Disease Control and Prevention (CDC), percent positivity in RSV diagnostic testing in the United States (US) failed to reach even 1% during the typical 2020–2021 season, compared to an average of 12–16% peak positivity in years prior[2]. However, the easing of NPIs in the summer of 2021 was accompanied by a broad, off-season increase in cases, roughly 32 weeks after the originally anticipated peak. Similar trends emerged worldwide, with delays in the expected RSV season ranging from

13 weeks in France[3] to up to 88 weeks in South Korea[4]. In some countries, including Japan[5] and the US, this was followed by a particularly large spike in cases in the 2022–2023 season that overwhelmed many pediatric hospital systems and highlighted the need for improved RSV-specific therapeutic options and preparedness[6].

RSV is one of the leading causes of acute respiratory tract infections worldwide, with particularly high disease impact on pediatric populations, older adults, and immunocompromised individuals[7]. Most children are initially infected with RSV before the age of 2, but rapidly waning immunity allows for regular reinfection throughout adulthood[8,9]. While most often resulting in mild respiratory illness in adults, several risk factors for more severe disease have been

[1]Division of Infectious Diseases, Northwestern University Feinberg School of Medicine, Chicago, IL 60611, USA. [2]Center for Pathogen Genomics and Microbial Evolution, Northwestern University Havey Institute for Global Health, Chicago, IL 60611, USA. [3]Northwestern Medicine Enterprise Data Warehouse, Northwestern University Feinberg School of Medicine, Chicago, IL 60611, USA. [4]Department of Infectious Diseases, University of California – Irvine, Orange, CA 92868, USA. [5]Division of Microbiology and Infectious Diseases (DMID), National Institute of Health, Rockville, MD 20852, USA. ✉e-mail: judd.hultquist@northwestern.edu

identified, including older age, immune dysfunction, and comorbid conditions such as chronic obstructive pulmonary disease and congestive heart failure[10]. Of the two major circulating subtypes of RSV, RSV-A has been more often associated with a higher risk of severe disease than RSV-B, though these studies have been largely limited to pediatric populations[11,12]. Nonetheless, a lack of widespread testing for RSV, especially in adults, suggests that the true incidence and burden of this virus remain underestimated[13]. However, in 2023 the Food and Drug Administration (FDA) approved two first-in-class RSV vaccines for older adults and maternal immunization. The first adjuvanted vaccine (Arexy) reduced the risk of RSV-lower respiratory tract infection (LRTI) in older adults by 94.1%[14]. The later unadjuvanted, bivalent vaccine (Abrysvo) reduced LRTI in older adults by 85.7%[15] and in infants from vaccinated pregnant individuals by 81.5% within 90 days after birth[16]. Likewise, the monoclonal antibody nirsevimab received FDA approval in July 2023 with a reduction of medically attended lower respiratory tract infection from 5.0% to 1.2% for children under 2 years of age[17]. While trials of these and other vaccines and monoclonal antibodies, most children and adults between 2 years of age and 60 have no current options approved to prevent RSV.

The use of these recently approved agents and the many ongoing clinical trials have increased the call for expanded genomic surveillance of RSV populations. While whole genome sequencing pipelines to monitor SARS-CoV-2 evolution proved to be critical to informing monoclonal antibody usage during the COVID-19 pandemic, RSV sequencing has remained much more limited[18]. While some diagnostic platforms distinguish between RSV subtypes[19], genotyping efforts are largely reliant on external surveillance/research teams and are typically limited to the sequencing of the attachment glycoprotein (G), which has the most variable open reading frame (ORF). This strategy precludes the detection of mutations in other parts of the genome, including those that arise in the viral fusion (F) protein, the main target of most vaccine and therapeutic strategies. However, efforts to standardize and expand global surveillance are currently underway by the World Health Organization Global RSV Surveillance Project[20]. Previous lack of prioritization in RSV genomic surveillance has had direct consequences on effective therapeutic design. For example, monoclonal antibody suptavumab failed to meet its endpoints in phase 3 clinical trial due to the unrecognized high frequency of naturally occurring escape mutations (F: S173L & F: L172Q) in the US at the time of study[21]. Current genomic surveillance efforts have identified increased genetic diversity in RSV following the COVID-19 pandemic, particularly in RSV-B, but it is unclear how these mutations might impact the efficacy of future therapeutics. Monitoring emerging resistance to nirsevimab has thus far been limited to computational modeling[22] and in vitro assays[23], with analyzed mutations limited to those that were prevalent before 2020 thus far.

Here, we present a retrospective cohort study that examined RSV epidemiology, clinical severity, and genetic diversity in Chicago, Illinois, US, in the years surrounding the COVID-19 pandemic. Our objective in this study was three-fold: (1) To elucidate shifts in RSV epidemiology during and after implementation of NPIs; (2) To test for associations between viral subtype and clinical outcomes in adult populations; and (3) To assess viral genetic diversity over time and its likely impact on upcoming treatment and vaccine efficacy.

## Results

### Alterations in RSV epidemiology in the United States following the COVID-19 pandemic

Before the COVID-19 pandemic, RSV cases in the US typically followed a regular seasonal pattern beginning in mid-October and lasting until early May, with slight regional variations[24]. Testing practices monitored through voluntary and passive reporting systems, such as the National Respiratory and Enteric Virus Surveillance System (NREVSS), largely echoed these trends with most tests administered between October and March[25]. To assess how RSV epidemiological trends changed in the US after the COVID-19 pandemic, we calculated 3-week rolling averages of diagnostic tests administered, RSV detections, and percent positivity from NREVSS data between July 2010 and April 2023 for all participating US laboratories (Fig. 1a). Diagnostic testing for RSV through the 2019–2020 season followed a regular seasonal pattern with the total number of tests administered peaking near 40,000 tests per three-week rolling average in later years. Detections and percent positivity rate followed similar seasonal trends, with most years peaking around 15 to 20% positivity during December or January. Testing increased as expected early in the 2020–2021 season, but detections and percent positivity remained near zero. Afterward, however, the number of tests administered remained high year-round with additional seasonal testing during the winter months pushing peak testing to over 140,000 tests per three weeks in early 2023. Cases likewise showed aberrant trends with a broad off-season peak during the summer and fall of 2021, followed by an early and dramatic increase in detections the following season in November of 2022. Despite the altered seasonality and testing patterns, percent positivity remained in line with historic trends, always peaking near 15 to 20%. Likewise, regional trends where the single-center study took place recapitulated the aberrant RSV behavior observed in national data (Supplementary Fig. 1).

We then leveraged hospitalization data from the RSV-Hospitalization Surveillance Network[26] to determine whether increased viral detections coincided with increased hospital admission rates. Hospitalization data for patients with RSV were obtained from 12 states (California, Colorado, Connecticut, Georgia, Maryland, Michigan, Minnesota, New Mexico, New York, Oregon, Tennessee, and Utah)

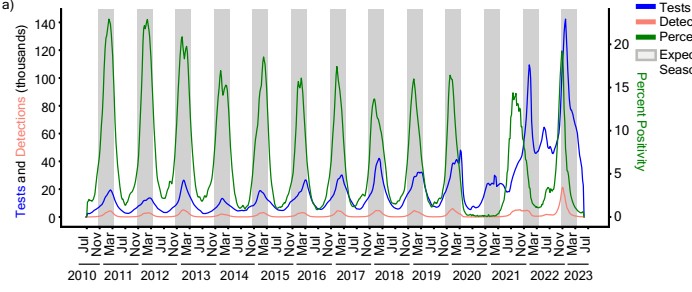
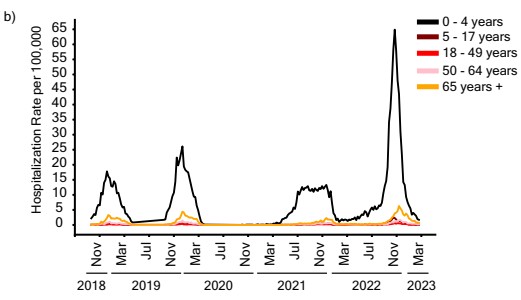

**Fig. 1 | Epidemiology of RSV in the United States from 2010 to 2023.**
**a** Epidemiology of RSV as represented by a 3-week rolling average of tests (blue), detections (pink), and percent positivity (green) between July 10th, 2010, and June 3rd, 2023 per data provided to the National Respiratory and Enteric Virus Surveillance System (NREVSS). Expected RSV seasonality (gray bars) is defined as early

November to late March. **b** Weekly Hospitalization rate per 100,000 individuals between October 13th, 2018 and March 11th, 2023 stratified by age groups: (i) 0–4 years (black), (ii) 5–17 years (maroon), (iii) 18–49 years (red) (iv) 50–64 years (pink), and (v) 65 years and older (orange) per data provided to the RSV-Associated Hospitalization Surveillance Network (RSV-NET).

participating in the Emerging Infections Program[27] or the Influenza Hospitalization Surveillance Program between July 2017 and April 2023[28]. Irrespective of age, admission rates were below 1 case per 100,000 in the 2020–2021 season, reemphasizing the lack of RSV circulation at that time (Fig. 1b). Infants 0 to 4 years of age displayed a broadened and off-season peak in hospitalizations in 2021 followed by a heightened peak in October 2022, approximately 3–4 times higher than in prior years. These trends are less apparent in other age groups, which showed a more typical seasonality, particularly for older adults aged 65 and older. Similarly, while hospitalizations increased for other age groups in the 2022–2023 season, the magnitude of the increase was not proportional to what was seen for infants. Taken together, these data confirm alterations in RSV epidemiological patterns following the COVID-19 pandemic but suggest these trends were largely driven by infants as opposed to older adults.

### Shifts in RSV testing platforms resulted in an incidental increase in outpatient detections

National RSV-associated hospitalizations did not increase proportionately with cases in the 2022–2023 season, suggesting an increase in less severe case detections. To explore the potential reasons behind this shift in proportionality, we performed a single-center retrospective cohort study of patients who tested positive for RSV at a Northwestern Medicine (NM) affiliated hospital or clinic between August 1st, 2009, and March 1st, 2023. Clinical data collated through NM's centralized repository system identified 8508 unique patient encounters over this timeframe, including 3416 adult and 5055 pediatric patients (Supplementary Fig. 2). Most NM patient encounters were geographically concentrated in the greater Chicago area (Cook County and the surrounding suburbs) (Fig. 2a). Within Cook County, encounters were aggregated in the north and far north regions of the city with lower representation in the southwest and far southwest regions. To verify that RSV behavior in Chicago was representative of broader regional and national trends, we obtained 3-week rolling averages of diagnostic tests administered, RSV detections, and percent positivity from the Chicago Department of Public Health (CDPH) from September 29th, 2019, to January 22nd, 2023 (Fig. 2b). As expected, these data confirmed aberrant seasonality and testing patterns citywide following the COVID-19 pandemic. Decreased RSV detections during the 2020–2021 season coincided with several city-wide COVID-19-related NPIs. These mitigation strategies included stay-at-home orders, indoor mask mandates, and remote learning for Chicago Public Schools as mandated by the Illinois Department of Public Health. Subsequent easing of these NPIs in early 2021 preceded an increase in off-season detections and percent positivity in the summer of 2021.

We next assessed patient outcomes (i.e., outpatient, hospitalized, ICU admission, and death) and diagnostic testing platform information for the 3416 adult patient encounters from our clinical dataset. Adult patient encounters increased gradually from the 2009–2010 season through the 2019–2020 season (Fig. 2c). These encounters were predominantly inpatients (>75%) until the 2015–2016 season when encounters became more balanced at around 50% for both outpatient and inpatient encounters (Fig. 2d). After only a single observed encounter in the 2020–2021 season during the COVID-19 pandemic, cases rebounded to near-normal levels in 2021–2022 before doubling in the 2022–2023 season. These latter seasons had increased outpatient encounters, with over 75% outpatient proportion during the 2022–2023 surge. Notably, outcomes among inpatients varied only slightly during the study period, with a consistent 25–40% of inpatients requiring intensive care or dying each year (Fig. 2d, bottom panel).

These data suggest that most adult cases in recent years were detected in the outpatient setting. Historically, RSV cases in the NM system were detected using either a broad, multiplexed, PCR-based respiratory panel or a similar PCR-based assay for the detection of RSV and influenza viruses (Fig. 2e). After the start of the COVID-19

pandemic, however, platforms became predominated by multiplex tests that included SARS-CoV-2, including an expanded broad respiratory panel assay, and a short triplex PCR assay for SARS-CoV-2, RSV, and influenza viruses. While the more expensive, multiplex panel was generally used for hospitalized and high-risk adults, the short triplex test was more frequently used in the outpatient setting[24]. The adoption of these platforms, particularly the short triplex assay, closely mirrored the increase in outpatient detections (Fig. 2e, Supplementary Fig. 3). Altogether, our patient and diagnostic data suggest that the transition to multiplexed platforms for the detection of SARS-CoV-2 led to an incidental increase in RSV detections among adults, particularly among patients not requiring inpatient care. As such, shifts in diagnostic testing platforms must be considered as an additional factor driving case counts in addition to virological and immunological behavior.

### RSV subtype A infection is associated with increased risk of ICU admission in hospitalized adults

When controlling for clinical and demographic confounders, a plurality of studies have found that RSV-A is associated with increased severity and worse patient outcomes compared to RSV-B[7,29,11,12]. Several other studies, however, have reported no difference by subtype, leading to an overall lack of consensus, likely due to the heterogeneity in inclusion criteria, timelines, severity metrics, and statistical methods[30–32]. Most of these studies were limited to pediatric populations and were conducted before the COVID-19 pandemic. To determine whether subtype is associated with clinical outcomes in our adult cohort, we subsampled our dataset to only include adult encounters with RSV subtyping information. Of the 3416 adult encounters, only 886 were tested on a diagnostic platform that could discern between subtypes, with largely no typing data available after 2017. To recover subtyping information, we collected 551 residual diagnostic nasopharyngeal swabs predominantly comprised of RSV-positive adults between December 18, 2017, and January 1, 2023, along with some incidental pediatric collections (Supplementary Fig. 2). RNA was extracted from each specimen and the RSV subtype was determined using an in-house quantitative reverse transcription PCR (qRT-PCR) assay. Merging the subtyping data from the 412 adult isolates with our clinical dataset, we recovered typing information for 1,437 patients, including outpatient ($n = 495$) and inpatient ($n = 942$) adults.

RSV-A and RSV-B were found to co-circulate each season, with year-to-year shifts in predominance (Fig. 3a). Following the COVID-19 pandemic, subtype predominance fluctuated more dramatically with the 2021–2022 season having a 91.9% RSV-B predominance, and the 2022–2023 season having an 80.8% RSV-A predominance. Inpatient adults with typing information [RSV-A ($n = 415$) or RSV-B ($n = 527$)] were highly representative of our overall inpatient cohort ($n = 1565$) (Table 1). The age distribution was similar among inpatients with RSV-A or RSV-B, skewing towards 60 years and older (Fig. 3b). Comorbidities were likewise similarly distributed among inpatients with different RSV subtypes, though patients with RSV-B reported slightly lower frequencies overall (Fig. 3c). We applied a Kaplan–Meier survival model to observe the probability of discharge among hospitalized patients and found no statistical difference ($p = 0.620$) between subtypes with a 50% discharge probability within 5 days from hospital admission for both A and B infections (Fig. 3d).

To determine whether an association exists between subtype and patient outcome, we used multivariable logistic regression to model subtype infection (i.e., RSV-A versus RSV-B) while controlling for race, ethnicity, sex, age at admission, the month of patient admission, comorbidity sum, hospitalization length of stay, intensive care unit (ICU) admission, and RSV-associated death. This model indicated that ICU admission [adjusted Odds Ratio (aOR) 0.68, 95% confidence interval (CI) 0.49–0.94] and comorbidity sum (aOR 0.92, 95% CI 0.85–0.99) were significantly associated with RSV-A infection (the reference subtype) as compared to RSV-B (Fig. 3e, Supplementary

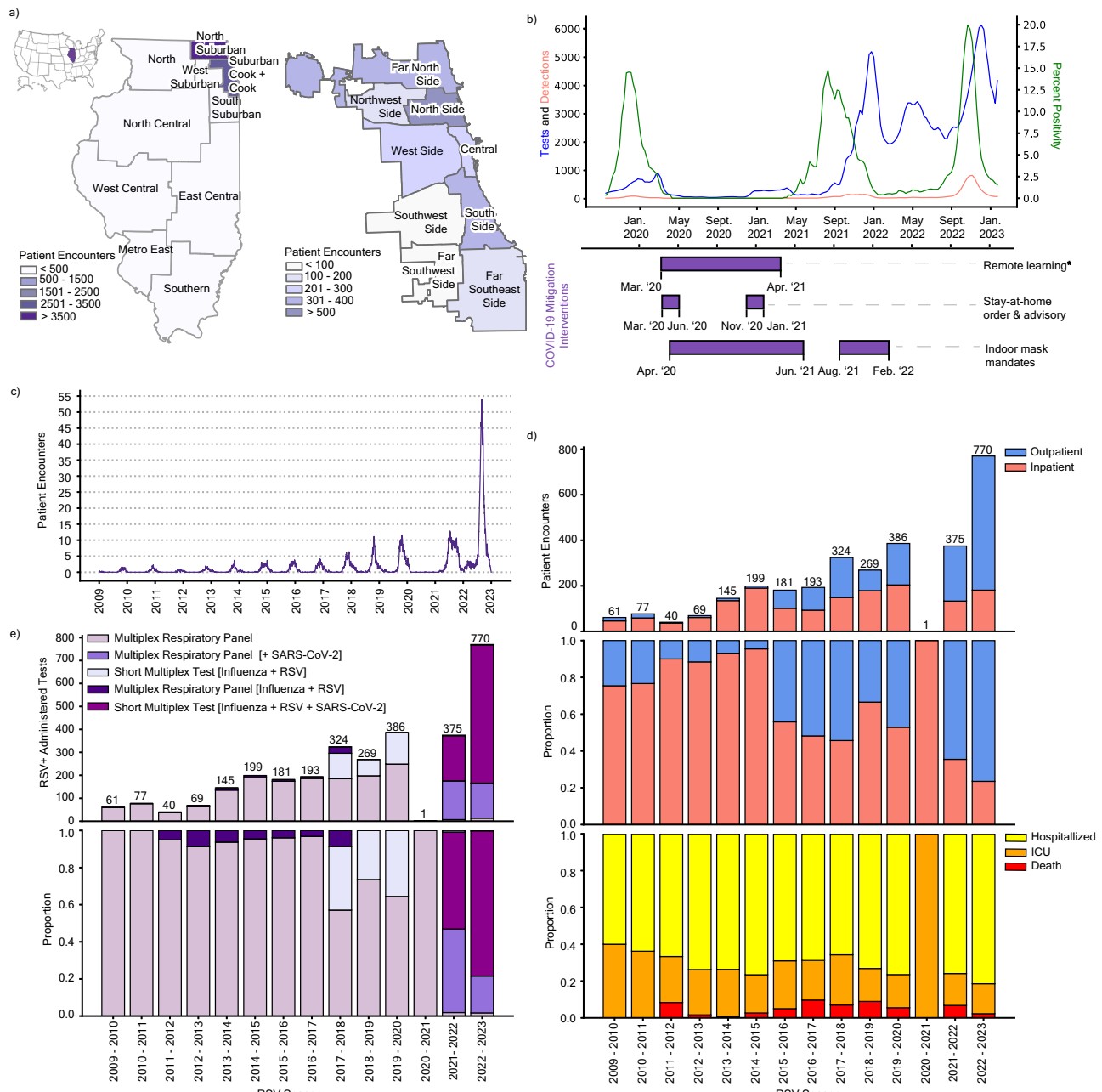

**Fig. 2 | RSV patient encounters reflect changes in testing platforms. a** RSV patient encounters between January 4th, 2009 and February 20th, 2023 sorted by zip code across the 11 Emergency Medical Service (EMS) regions in Illinois (left; as defined by the Illinois Department of Public Health) and the 9 community regions in the city of Chicago (right; as defined by regional census data). The location of Illinois in the US is annotated in purple (top left). **b** 3-week rolling average of RSV tests administered (blue), detections (pink), and percent positivity (green) data from the Chicago Department of Public Health (CDPH) from September 29th, 2019 to January 22nd, 2023. The timeline of COVID-19 mitigation measures implemented in Chicago is represented in purple bars below. *The end of remote learning for

Chicago Public Schools (CPS) was staged with K-8th grade return to in-person classes on March 1st, 2021, and 8th–12th grade return on April 19th, 2021. **c** Weekly rolling average of adult patient encounters in NM-affiliated institutions from January 4th, 2009 to February 20th, 2023. **d** Absolute counts (top) and proportion (middle) of NM-affiliated adult RSV patient encounters grouped by outpatients (blue) and inpatients (pink). Inpatients are further stratified by the proportion of those hospitalized without ICU admission (yellow), hospitalized with ICU admission (orange), and death (red) per RSV season (bottom). **e** Absolute counts (top) and proportion (bottom) of NM-affiliated adult patient encounters with a positive RSV test grouped by the diagnostic platform in each season (February to January).

Fig. 4). Likewise, we observed an overall increased proportion of ICU admissions in our RSV-A infected cohort (32.8%) in comparison to the RSV-B cohort (27.1%) (Fig. 3f). These same trends were observed when excluding inpatient encounters after 2020, suggesting that these associations were not confounded by COVID-19 pandemic-associated factors (Supplementary Fig. 4). Likewise, similar relationships were observed when using multivariable logistic regression to model ICU admission explicitly (Supplementary Fig. 5). Altogether, our modeling

data are indicative of more severe outcomes in hospitalized adults with RSV-A compared to RSV-B and corroborate previous reports in pediatric populations.

## Shift in RSV-B population structure following the COVID-19 pandemic

The RSV resurgence superseding the low incidence in the 2020–2021 season has been reported to be from previously existing RSV lineages

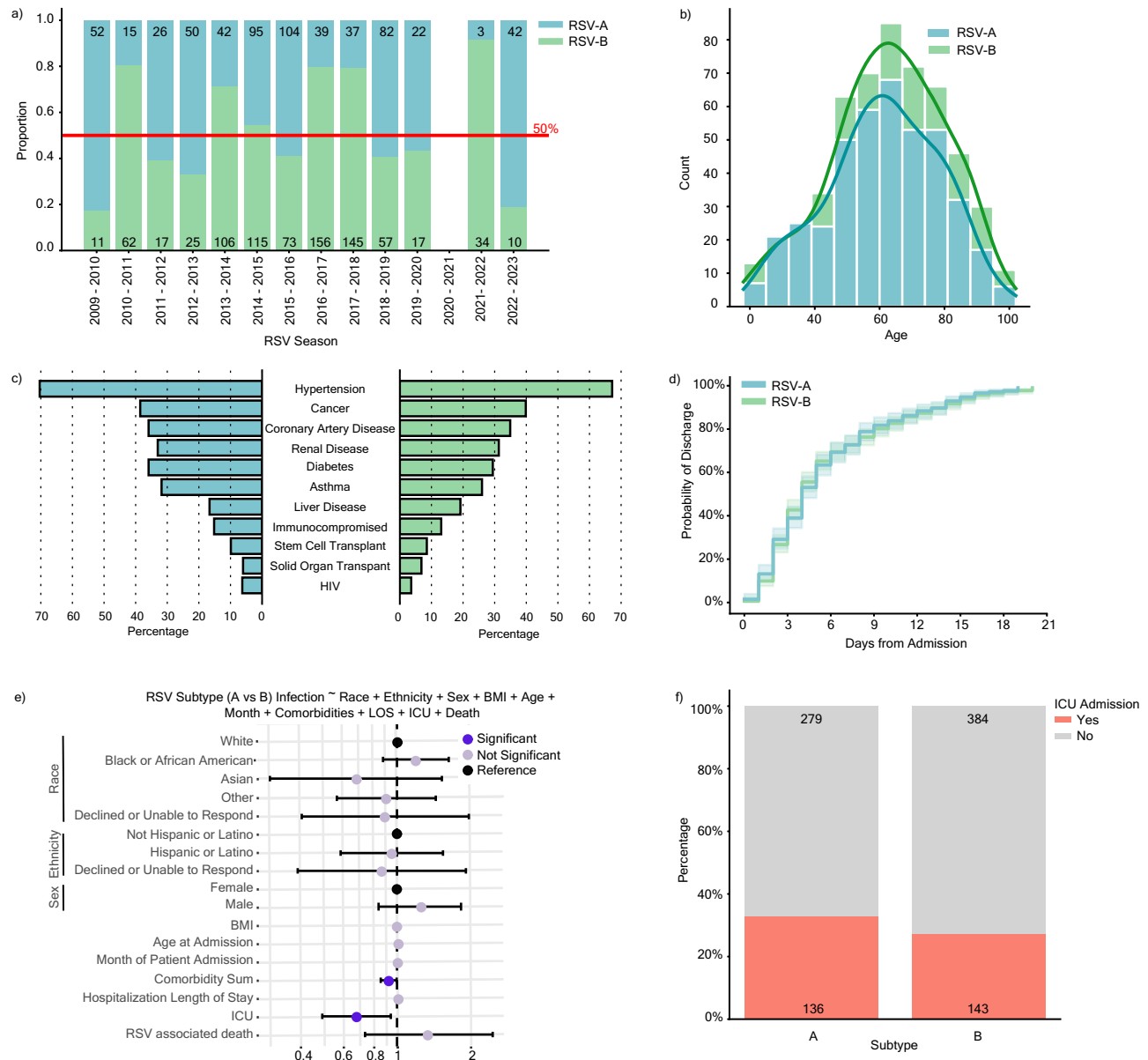

**Fig. 3 | RSV-A is associated with higher risk of ICU admission. a** Distribution of RSV subtype by season from April 2, 2009 to March 1, 2023 among 1437 outpatient ($n = 495$) and inpatient ($n = 942$) adults with typing information as reported by diagnostic platforms or through in-house PCR-based typing [RSV-A in blue ($n = 609$), RSV-B in green ($n = 828$)]. **b** Histogram of the age distribution of adult inpatients stratified by subtype ($n = 415$ RSV-A and $n = 527$ RSV-B). **c** Percent of adult inpatients with the indicated comorbid condition stratified by subtype. **d** Kaplan–Meier plot depicting the probability of discharge within 20 days of an RSV-associated hospitalization stratified by subtype ($n = 377$ RSV-A and $n = 489$ RSV-B). Inpatients with hospital stays longer than 20 days are not shown ($n = 76$).

There is no statistical difference between subtypes by log-rank test ($p = 0.620$). **e** Odds ratio plot with 95% confidence intervals (CI) as calculated by a multivariable logistic regression model with RSV-A (reference) or RSV-B infection as the outcome variable for hospitalized adults ($n = 942$). Significant features ($p < 0.05$) are highlighted in dark purple (comorbidity sum; 0.025, ICU admission; 0.018), insignificant features ($p >= 0.05$) are shown in light purple, and reference categories for categorical variables are shown in black (refer to $p$ values in Supplementary Fig. 4A). **f** Percentage and absolute count of inpatient cohort experiencing ICU admission (yes, orange, $n = 279$) and either outpatient or hospitalization admission (no, gray, $n = 643$).

that now predominate in the US[33]. To determine if the genetic diversity of RSV in Chicago changed throughout the COVID-19 pandemic, we expanded the use of the 551 residual diagnostic nasopharyngeal swabs used for typing for viral whole genome sequencing (WGS, Supplementary Fig. 2). Specifically, viral genomic material from the collected isolates was extracted, reverse transcribed, amplified using an overlapping fragment approach spanning the whole genome, and sequenced using the Illumina platform. In total, we obtained 218 whole genome sequences (110 subtype A and 108 subtype B) with a minimum of 90% coverage across the genome. Sequences were given a clade

designation using both Nextstrain and G clades nomenclature[34] via Nextclade and were deposited in the National Center for Biotechnology Information (NCBI) GenBank (Supplementary Table 1). Finally, maximum-likelihood (ML) phylogenetic trees were generated for all RSV-A and RSV-B sequences.

Between 2017 and 2023, the RSV-A and RSV-B populations in Chicago were exclusively represented by existing clades GA2.3.5 and GB5.0.5a, respectively (Fig. 4a, b). The RSV-A population structure remained largely unchanged after 2020, showing evidence of multiple lineage expansions (Fig. 4a), consistent with prior reports from

**Table 1 | Demographic and clinical features of RSV-positive adult inpatients**

| Category | Overall (n = 1565) | RSV-A (n = 415) | RSV-B (n = 527) |
|---|---|---|---|
| **Age at Admission, median (SD)** | | | |
| | 64.0 (18.1) | 62.0 (17.5) | 63.0 (17.7) |
| **Race, n (%)** | | | |
| White | 876 (56.0) | 213 (51.3) | 276 (52.4) |
| Black or African American | 457 (29.2) | 124 (29.9) | 169 (32.1) |
| Asian | 56 (3.6) | 14 (3.4) | 12 (2.3) |
| Other | 127 (8.1) | 46 (11.1) | 51 (9.7) |
| Declined or Unable to Answer | 49 (3.1) | 18 (4.3) | 19 (3.6) |
| **Ethnicity, n (%)** | | | |
| Hispanic or Latino | 167 (10.7) | 43 (10.4) | 46 (8.7) |
| Not Hispanic or Latino | 1344 (85.9) | 354 (85.3) | 463 (87.9) |
| Declined or Unable to Respond | 54 (3.5) | 18 (4.3) | 18 (3.4) |
| **Gender, n (%)** | | | |
| Female | 898 (57.4) | 247 (59.5) | 294 (55.8) |
| Male | 667 (42.6) | 168 (40.5) | 233 (44.2) |
| **Body Mass Index (BMI), mean (SD)** | | | |
| | 29.2 (9.8) | 29.8 (9.5) | 28.9 (8.6) |
| **Intensive Care Unit (ICU) Admission, n (%)** | | | |
| | 415 (26.5) | 136 (32.8) | 143 (27.1) |
| **RSV Attributable Death, n (%)** | | | |
| | 93 (5.9) | 21 (5.1) | 33 (6.3) |
| **RSV Subtype, n (%)** | | | |
| A | 415 (26.5) | 415 (100.0) | 0 (0.0) |
| B | 527 (33.7) | 0 (0.0) | 527 (100.0) |
| Unknown | 623 (39.8) | 0 (0.0) | 0 (0.0) |
| **Comorbidities, n (%)[a]** | | | |
| Solid Organ Transplant | 94 (6.0) | 25 (6.0) | 36 (6.8) |
| Stem Cell Transplant | 134 (8.6) | 41 (9.9) | 45 (8.5) |
| Liver Disease | 279 (17.8) | 69 (16.6) | 101 (19.2) |
| Diabetes | 512 (32.7) | 149 (35.9) | 155 (29.4) |
| Renal Disease | 479 (30.6) | 137 (33.0) | 165 (31.3) |
| Cancer | 610 (39.0) | 160 (38.6) | 210 (39.8) |
| Asthma | 432 (27.6) | 132 (31.8) | 137 (26.0) |
| Coronary Artery Disease | 539 (34.4) | 149 (35.9) | 184 (34.9) |
| Hypertension | 1071 (68.4) | 292 (70.4) | 354 (67.2) |
| Immunocompromised | 201 (12.8) | 63 (15.2) | 69 (13.1) |
| Human Immunodeficiency Virus (HIV) | 74 (4.7) | 26 (6.3) | 19 (3.6) |
| **Comorbidity Sum, mean (SD)** | | | |
| | 2.8 (1.9) | 3.0 (1.8) | 2.8 (1.8) |
| **Smoking Status Upon Admission, n (%)** | | | |
| Current | 133 (8.5) | 25 (6.0) | 38 (7.2) |
| Former | 514 (32.8) | 123 (29.6) | 170 (32.3) |
| Never | 635 (40.6) | 174 (41.9) | 194 (36.8) |
| Unknown | 283 (18.1) | 93 (22.4) | 125 (23.7) |

Continuous variables are presented as median or mean and standard deviation (SD); categorical variables are presented as n (%). The subset of total inpatients (n = 1565) with known genotyping information for RSV-A (n = 415) and RSV-B (n = 527) are presented in adjacent columns for comparison.

[a]Comorbidities are not mutually exclusive variables.

Arizona[35], Washington State[36], and Massachusetts[33]. Conversely, RSV-B exhibited monophyletic clustering with strong support (100% using Shimodaira–Hasegawa approximate likelihood-ratio test [aLRT] and 100% with bootstrap) after 2020 (Fig. 4b), suggestive of population bottlenecking caused by either genetic drift or positive selection. The defining mutations for this monophyletic cluster were mainly in glycoprotein (I252T, I268T, S275P, P214S, P221L, T310I, and S100G) and the fusion protein (S190N, S211N, S389P). To better determine whether this cluster is specific to Chicago and its surrounding suburbs, we pulled 340 publicly available RSV-A and 113 publicly available RSV-B whole genome sequences from the US and repeated our phylogenetic analysis. Again, RSV-A sequences after 2020 were broadly representative of GA.2.3.5 diversity in the US before that year (Fig. 4c). However, RSV-B sequences after 2020 formed a monophyletic cluster composed of our sequences alongside those from Washington and Massachusetts (Fig. 4d). Taken together, these results suggest a potential founder effect in the RSV-B population during the COVID-19 pandemic, which resulted in a more homogenous population structure in the US post-pandemic.

### Convergent evolution of mutations in the RSV-B fusion protein antigenic sites

While the US experienced an RSV-B dominant season after the easing of NPIs in 2021–2022, potentially contributing to the observed monophyletic outgrowth, several other countries experienced an RSV-A dominant season[37–39]. To determine whether this monophyletic cluster of RSV-B was unique to the US, we generated a temporal ML phylogenetic analysis using all publicly available RSV-B whole genome sequences collected between 1957 and 2023 (n = 723) (Fig. 5a). This analysis revealed genetically and geographically distinct clusters of RSV-B GB5.0.5a isolates arising in the 2021–2022 season (i.e., unique clusters were observed in Australia, France, Japan, the US, etc.). Similar temporal ML analyses of unique and complete RSV-B G (n = 983) and F (n = 2067) sequences showed the monophyletic divergent lineage in RSV-B after 2020 (Supplementary Fig. 6).

Subsequently, we used a Bayesian approach to characterize the global population dynamics of RSV-B and compared it with RSV-A (Supplementary Fig. 7) to identify the possible origin and the extent of the expansion of the detected circulating monophyletic lineage in the US. Overall, the population dynamics observed using this approach depicted a temporal evolution of the effective population size for both RSV-A and RSV-B concordant with the epidemiological patterns previously described. More importantly, these analyses confirmed strong statistical support for a monophyletic origin of most of the currently circulating RSV-B viruses in the US with a most likely local origin [ancestral US location probability >0.99]. Using this approach, we also estimated a time to the most recent common ancestor (TMRCA) for this cluster to be June 9, 2018 [95% Highest Posterior Density (HPD) interval: November 23, 2017–December 28, 2018] with the next MRCA for this cluster also being identified close to 2 NM sequences sampled between March and April 2018 that constituted a minority lineage at that time. This is indicative of possible selective processes involved in the population bottleneck observed due to the unlikeliness of random drift to a minority lineage instead of within the most prevalent lineages. Although most of the sequences that constituted this cluster were collected in the US, 4 sequences from Austria from NCBI collected during 2022 also clustered within this lineage, indicating ongoing expansion outside the US. On the other hand, RSV-A showed the appearance of multiple unrelated lineages after the period of the SARS-CoV-2 pandemic discarding any possible sampling bias leading to the identification of the dominant monophyletic US cluster in RSV-B.

To further characterize the mutations in the F ORF that define currently circulating isolates and test their possible selective advantage, we generated an ancestral sequence of the most recent common ancestor (MRCA). When comparing the MRCA sequence to all publicly available RSV-B sequences available after 2020 (n = 225), we observed an enrichment of mutations on the heptad-rich A domain (HRA), which is necessary for F protein conformational change and viral-host membrane fusion (Fig. 5b, left)[40]. When stratifying these data by country, three mutations in the HRA domain (K191R, I206M, and Q209R) were particularly enriched, predominating in 12 of the 15

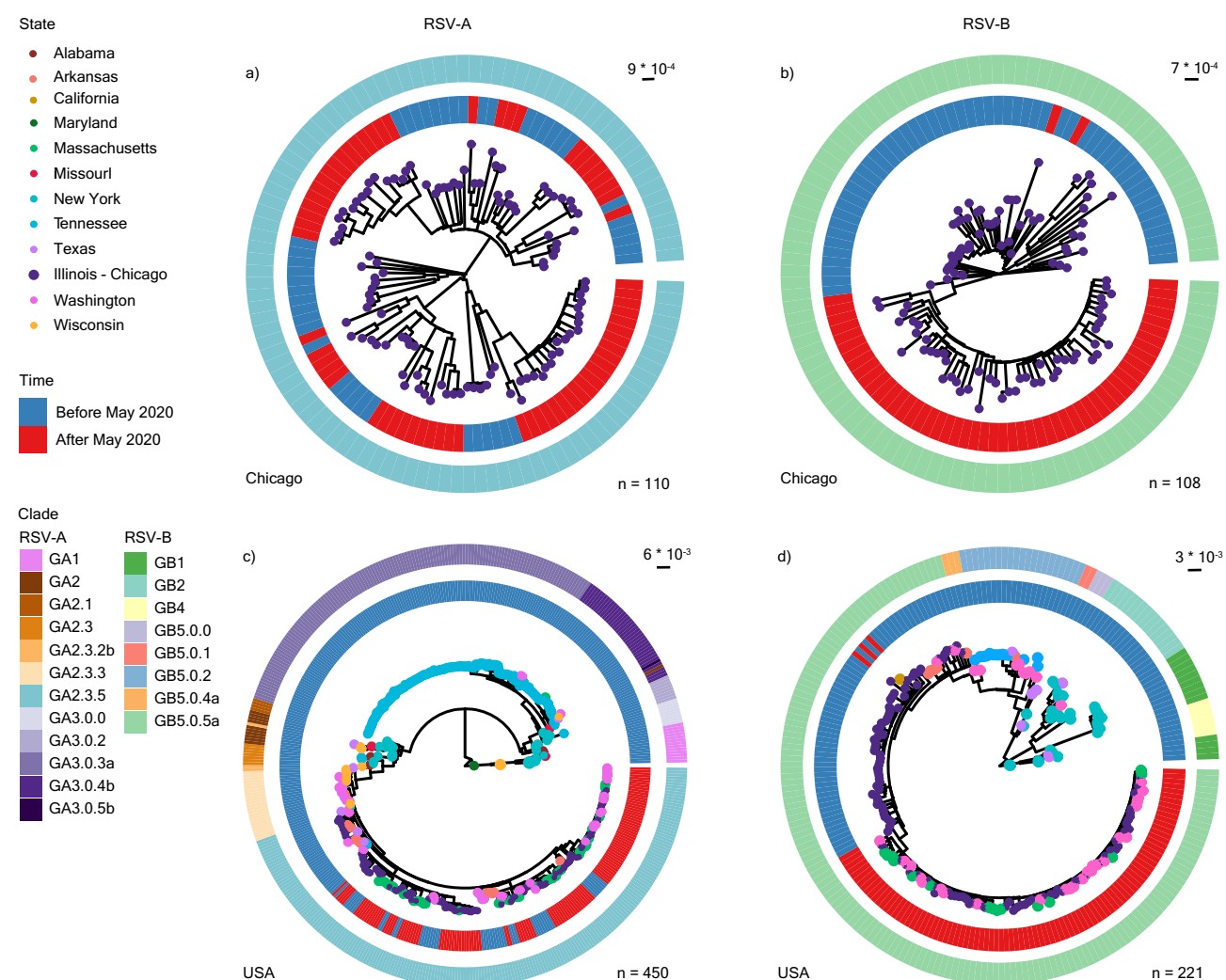

**Fig. 4 | Phylogenetic analysis of RSV-A and -B whole genome sequences.**
**a** Maximum likelihood (ML) phylogenetic analysis of RSV-A whole genome sequences from NM-affiliated institutions in Chicago, Illinois from 2018 to 2022 ($n = 110$). **b** Equivalent ML phylogenetic analyses of RSV-B whole genome sequences from NM-affiliated institutions in Chicago, Illinois from 2017 to 2022 ($n = 108$). **c** ML phylogenetic analysis of RSV-A whole genome sequences from NM-affiliated institutions in Chicago, Illinois with publicly available genomes from the United States

(US) collected from 1956 to 2022 (as of March 1st, 2023) ($n = 450$). **d** ML phylogenetic analysis of RSV-B whole genome sequences from NM-affiliated institutions in Chicago, Illinois with publicly available genomes from the US collected from 1979 to 2022, $n = 221$. For all trees, branch tips are colored by state of origin, the inner ring is colored by time (blue = before May 2020, red = after May 2020), and the outer ring is colored by clade as designated by Nextclade v2.14.1 (refer to Supplementary Table 1).

countries with available sequencing data after 2020 (Fig. 5b, right). While these mutations were all in the majority before 2020, several minority mutations also arose in multiple countries. For example, S190N and S211N, also in the HRA domain, arose to predominance in the US, Austria, and Canada, despite being present in fewer than 2% of sequences before 2020. Not all mutations were localized to the HRA domain, with, for example, P312H and S389P in Domain I which arose in several countries. Notably, most of these mutations lie either in or are flanking antigenic sites Ø, I, and V (Fig. 5c).

Given the position of several of these mutations in or near antigenic regions, we next sought to determine if any of these mutations were under positive selection. Each RSV-B ORF was analyzed independently for episodic diversifying selection using the Mixed Effects Model of Evolution (MEME) method using all available sequence data from 1957 to 2023. Most ORFs showed no evidence of positive selection over this time frame, including NS1, NS2, N, M, P, and SH. M2-1 (T188), M2-2 (S26, H50), and L (S69, S102, N500, I1989) each had positions showing evidence of episodic selection, but these mutations remained low frequency and none are currently circulating (Supplementary Fig. 8). However, both F (Fig. 5d, left) and G (Supplementary

Fig. 8) showed high levels of episodic positive selection at positions 7 and 20, respectively. The 252 position in G is identified to undergo diversifying selection and is a defining mutation site in the monophyletic RSV-B cluster. However, no sites undergoing positive selection in F are fully represented in the cluster, with the F12L mutation only being represented in less than 5% of that cluster. As expected, S173L, a known resistance mutation against Suptavumab in antigenic region V, is now fixed in the RSV-B population. Expanding this analysis to include pervasive positive selection using the Fast Unconstrained Bayesian AppRoximation (FUBAR) method, two sites in the F signal peptide (I5 and F12) are additionally identified (Fig. 5d). If we constrain these analyses to clade GB5.0.5a in the last 5 years, only three mutations are identified to still be under episodic (D548) and pervasive (I5 and F12) positive selection (Fig. 5d, right). While the predominant mutations arising after 2020 are not found to be under statistically significant positive selection, several have high mean posterior difference values between nonsynonymous and synonymous mutations.

Lastly, we quantified the mutational frequency within the binding sites for the monoclonal antibody therapeutics nirsevimab (Fig. 5e),

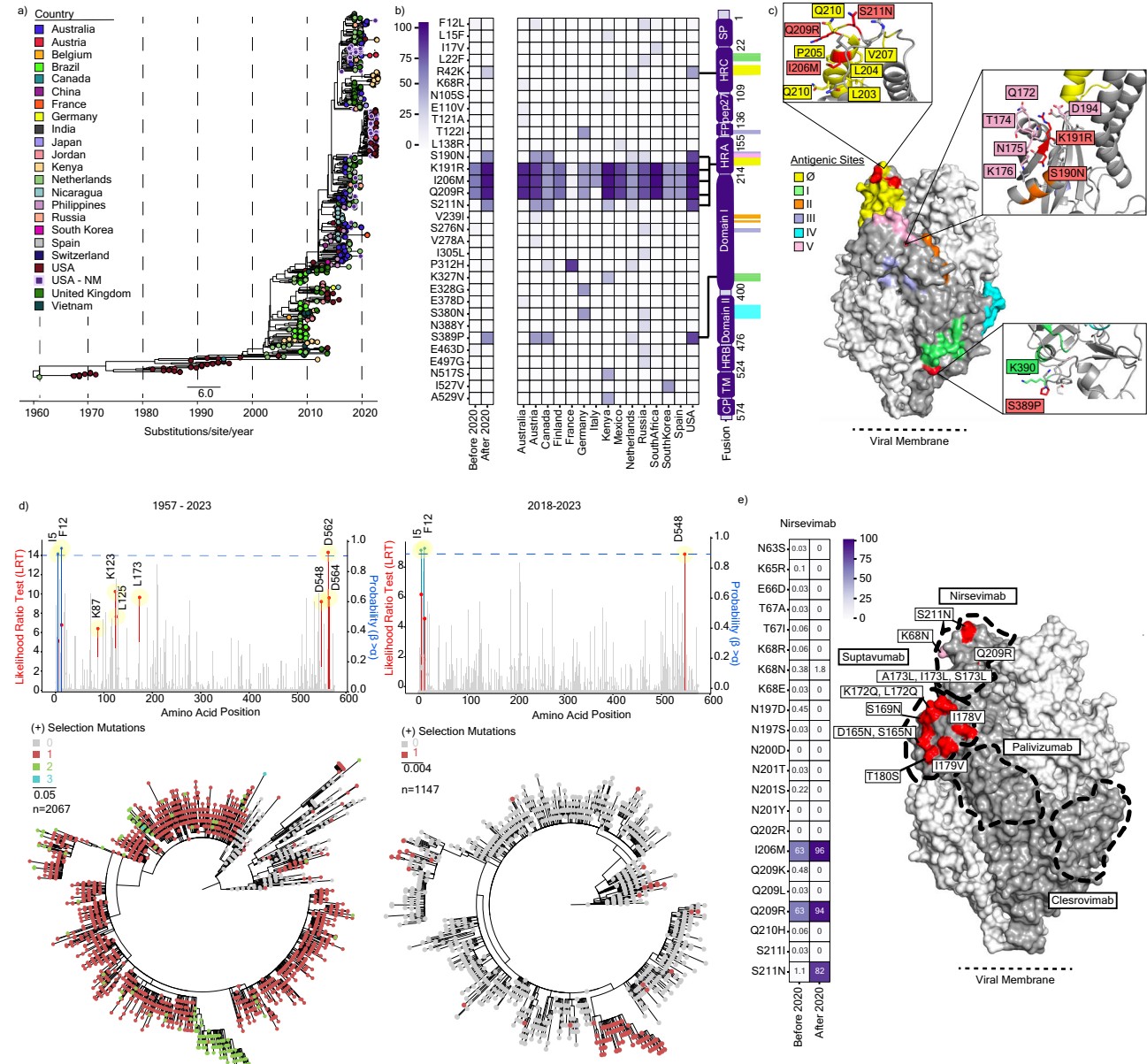

**Fig. 5 | Globally circulating RSV-B mutations in fusion protein. a** Maximum likelihood (ML) phylogenetic temporal analysis of globally circulating RSV-B whole genome sequences collected from 1957 to 2023 (as of October 10, 2023) (*n* = 723). Branch tips are colored by country of origin, and scale units are substitutions/site/ year. **b** Heatmap of RSV-B Fusion mutational frequency among sequences sampled before and after May 1, 2020 relative to the GB5.0.5a most recent common ancestor (MRCA, left). Isolates sampled after May 1, 2020, are further stratified by country (*n* = 15, middle). A schematic of F domains with antigenic site positions highlighted is shown on the right. **c** Structure homology modeling of RSV-B prefusion F from the NMH/US cluster using template PDB:6QOS via SWISS-MODEL. Antigenic sites (yellow = Ø, green = I, orange = II, purple = III, cyan = IV, pink = V) and mutations defining the NMH/US cluster (red) are annotated on the trimer structure. Side chain residues within 5 angstroms of the three US cluster-defining mutations (S190N, S211N, S389P) are labeled in the insets. **d** Overlayed lollipop and bar plots for

episodic and pervasive positive selection analysis, respectively, of unique RSV-B F open reading frames (ORFs) from 1957 to 2023 (*n* = 2067, right) and from 2018 to 2023 (*n* = 1147, left). Lollipop plots depict the likelihood ratio test (LRT) for episodic positive selection by a mixed effects model of evolution (MEME); significant positions are highlighted in red (*p* < 0.05). Overlaid bar charts depict the posterior probability of pervasive positive selection by Fast Unconstrained Bayesian AppRoximation (FUBAR); significant positions at greater than 90% are highlighted in blue. Corresponding ML phylogenetic trees (bottom) display the number of positions undergoing positive selection in a given branch. **e** Heatmap displaying mutational frequencies at positions within the Nirsevimab binding site before and after May 1, 2020 (left). Mutational frequency by position after May 1, 2020, is overlaid on the RSV-B prefusion F homology model (right). Proposed monoclonal antibody binding sites for Nirsevimab, Suptavumab, Clesrovimab, and Palivizumab are outlined.

suptavumab, clesrovimab, and palivizumab (Supplementary Fig. 9) before and after 2020 (Fig. 5e). As expected, current RSV-B isolates contain a high frequency of mutations across the Suptavumab binding site both before and after 2020, with treatment-resistant mutations (i.e., S173L) observed in all isolates. However, several mutations in the Nirsevimab binding site, including I206M, Q209R, and S211N, have substantially increased in frequency since 2020 with minor increases

observed at other positions (i.e., K68N). Overall, mutations in the Clesrovimab and Palivizumab binding sites remain low.

## Discussion
In this single-center study, we leveraged a 14-year clinical dataset and a matched biobank of residual diagnostic specimens to assess changes in RSV epidemiology, clinical severity, and genetic diversity in the

years surrounding the COVID-19 pandemic. We demonstrated that changes in diagnostic testing, particularly multiplexing of SARS-CoV-2 and RSV diagnostics, were associated with increased outpatient detections among adults, a factor that might contribute to the increased case counts observed in recent seasons. We further found that RSV-A was associated with a higher risk of ICU admission among hospitalized adults, consistent with what has been previously reported in pediatric populations. Finally, we observed a shift in the RSV-B population structure to geographically distinct monophyletic clusters in 2021–2022, likely a result of population bottlenecking during the widespread use of NPIs to mitigate the spread of COVID-19. Despite the independent emergence of differing RSV-B clusters, the convergent evolution of several mutations in the antigenic regions of F suggests potential alterations to therapeutic efficacy profiles that should be considered as new treatment options progress through clinical trials.

Data from within our hospital system reflected local and national trends in RSV surveillance with altered seasonality in 2021–2022 and a surge of detections in the 2022–2023 season. The resurgence of RSV in Chicago in the summer of 2021 closely followed the return to in-person learning in April and the recension of the indoor mask mandate in June, NPIs associated with delayed RSV resurgence in other countries[41]. Reduced pathogen exposure on account of the widespread use of NPIs during the COVID-19 pandemic is subsequently theorized to have produced an "immunity debt"[42] with a larger susceptible population resulting in larger surges of infection. Increased transmission and skewed age distributions in the 2021-2023 RSV seasons support this association between increased incidence and waned population immunity in infants. For example, a recent single-center study in the US reported a shift in median age from 11.0 months before 2020 to 18.5 months in 2022–2023[43]. This has likewise been reported in other countries where a decrease in anti-RSV IgG antibody concentrations was correlated with a shift in age distribution to older children[44]. On the other hand, adults likely retained immunity from prior RSV exposures despite a decreased probability of exposure during the COVID-19 pandemic. Consequently, while data from RSV-NET showed a coordinated increase in pediatric (0–4 years) hospitalizations and detections, adult hospitalizations did not increase to the same extent[45] and overall percent positivity remained comparable to prior seasons. According to our data, this can be partially explained by an increase in outpatient detections, which is strongly associated with a shift in the diagnostic testing platforms being used in our hospital system. The increased use of pan-viral molecular diagnostics platforms may increase the number of incidental detections that must be considered when interpreting epidemiological trends. This observed increase in molecular diagnostic usage could likewise be attributed to shifts in practice and societal norms regarding testing. Namely, changes in institutional/medical practices to order diagnostics, widespread availability of triplex testing, and behavioral shifts to preferentially seek COVID-19 testing during the pandemic are examples of societal factors that potentially contribute to increased RSV detection in recent years.

Prior studies in pediatric populations have shown that RSV-A infections have more severe clinical manifestations and worse patient outcomes compared to RSV-B[12,46]. While fewer studies have been done in adult populations, they likewise have indicated higher RSV-A virulence as measured by clinical severity scores[47]. Following the resurgence of RSV post-2020, several multi-center studies observed increased infant hospitalization rates with unchanged clinical severity trends[48,49], though no such contemporary studies have yet been reported in adults. Our data support the theory that worse clinical outcomes among hospitalized adults occur with RSV-A infection compared to RSV-B. While our dataset post-2020 is too sparse for a stand-alone model, our regression analyses illustrate that the inclusion or exclusion of adult inpatients sampled after 2020 does not contribute to associated risks of outcome and subtype infection. This suggests there is no change in relative severity between RSV-A and

RSV-B following the COVID-19 pandemic, but larger, multi-center studies with more patients are still required.

Our phylogenetic analyses revealed a shift in RSV-B population structure post-2020, with multiple monophyletic clusters arising in distinct geographic locations in the 2021–2022 season. The isolates in Chicago appear closely related to those previously reported in Washington state[36] and Massachusetts[33], suggestive of a US-specific cluster. This pattern of divergence is most consistent with a genetic bottleneck event, likely resulting from the NPIs introduced in 2020 to stop the spread of SARS-CoV-2. However, RSV-A global sequences exhibited polyphyletic clustering in 2022, suggesting that the transmission patterns between these two subtypes differ despite their co-circulation in the 2019–2020 season. This is further supported by the lack of monophyletic clustering of RSV-A even in regions where this subtype was dominant after viral reemergence in 2021[50]. Alternately, it is possible that the bottlenecking exacerbated an already ongoing transition in the RSV-B population. Viral populations with a mildly beneficial fitness advantage that are expanding in a low-competing environment may become rapidly predominant. The seemingly independent selection of similar mutations in F domains, several of which existed to some degree before the pandemic, suggest some convergent evolution at these sites, though the impact of these changes on viral entry, transmission, and immune evasion remains to be elucidated.

The RSV sequences reported here add to a growing body of publicly available RSV genomic data, which will improve our ability to monitor for shifts in viral evolution and emerging resistance mutations. This is particularly critical as a slate of new interventions gain FDA approval or begin to go through clinical trials. Prior literature on nirsevimab neutralization escape mutations showed low prevalence K68N (0.32%) and S211N (1.14%) and a global increase in Q209R (68.25%) and I206M (68.9%) between 2015 and 2021[51,52]. Although neutralization potency is reduced marginally by I206M (5.0-fold), Q209R (0.5-fold), and S211N (1.2-fold), the K68N mutation exhibits 29.9-fold reduced susceptibility[51]. The prevalence of each of these mutations has increased notably, especially S211N, which is now observed in 82% of sequences. Perhaps even more concerning, circulating RSV-B K68N mutations after May 2020 increased from <1% prevalence to 1.8% in Australia and the United States, suggestive of a growing population of strongly resistant isolates. Fortunately, other monoclonal antibody epitopes for either FDA-approved (palivizumab), investigational, or unapproved therapeutics (clesrovimab and suptavumab) do not exhibit a large shift in variation. However, our data show the emergence of several other mutations in the HRD A domain flanking antigenic sites Ø and V, which may cause issues for other antibodies targeting these sites. Altogether, the shift in RSV-B population structure and the emergence of nirsevimab-escape mutants strongly argue for further surveillance, particularly in regions committed to rolling out monoclonal antibody treatments in the 2023 RSV season.

Several limitations exist within our study design and analysis despite its contemporary insights into RSV epidemiology. For example, single-center cohort studies run the risk of not being representative of results in other medical institutions because of sampling methods, inclusion criteria, and model design. Despite this, the epidemiological data in Chicago appears indicative of US trends. Similarly, the incorporated clinical metadata was filtered to a smaller subset to focus on inpatient adults for our models. Consequently, removing outpatients underpowers our modeling of the association with adult clinical severity and RSV subtype post-2020. Again, this observed shift in hospitalization-to-outpatient ratios shows the impact of diagnostic shifts on association studies. Lastly, the scarcity of publicly available genomes results in inherent under-sampling bias, impacting the elucidation of phylogenetic and dynamic patterns.

Overall, this study characterized the molecular epidemiology of RSV in Chicago spanning the COVID-19 pandemic. Specifically, we sought to: (1) elucidate shifts in RSV epidemiology during and after

implementation of NPIs; (2) test for associations between viral subtype and clinical outcomes in adult populations; and (3) assess viral genetic diversity over time and its likely impact on upcoming treatment and vaccine efficacy. This multifaceted work demonstrates (1) the strengths that lie in utilizing epidemiology to inform clinical care, public health interventions, and molecular virology, and (2) the existing gaps in molecular surveillance in under-sample regions that impede our understanding of RSV evolution. Our findings also highlight the merit of currently developing virological techniques for future studies to establish causal relationships with identified mutations and functions[53]. Overall, efforts to establish and strengthen our understanding of RSV evolution are crucial, as approaching pharmaceutical interventions may further drive diversification and selection in the viral population with global implications.

## Methods

### Ethics approval
The collection of RSV-positive residual diagnostic specimens for viral whole genome sequencing was approved by the Northwestern University Institutional Review Board through IRB #STU00212260 and #STU00206850. Clinical and demographic data is stored on a secure Research Electronic Data Capture (REDCap) server in compliance with HIPAA regulations following IRB #STU00207123. Data were de-identified and linked to sequencing data for analysis using a non-descriptive, unique identifier.

### Public data extraction
Publicly available whole genome sequences were extracted from the NIH NCBI nucleotide database. Accession number and metadata information are available at: (https://github.com/erg6437/RSV-Molecular-Epidemiology). Case count, detections, and tests (antigen and PCR-based diagnostics) administered data were extracted from the Respiratory Syncytial Virus Laboratory Data (NREVSS) Surveillance and Analytics Team (https://data.cdc.gov/Laboratory-Surveillance/Respiratory-Syncytial-Virus-Laboratory-Data-NREVSS/52kb-ccu2).
Data extending beyond from June 27th, 2020 to June 3rd 2023 were not available in this dataset, so the data were consequently accessed in the CDC NREVSS National Trends (https://www.cdc.gov/surveillance/nrevss/rsv/natl-trend.html). CDPH testing, case count, and percent positivity data were provided by the Vaccine-Preventable Disease Surveillance Program.

### Northwestern medicine data extraction and specimen collection
Demographic and clinical metadata from RSV+ patient encounters were extracted from NM's enterprise data warehouse (EDW). A biobank of RSV diagnostic specimens was formed under the Center of Pathogen Genomics and Microbial Evolution (CPGME) Center at Northwestern University. Residual diagnostic nasopharyngeal swabs from patients with a confirmed positive RSV infection in the Northwestern Medicine healthcare system were collected from December 17, 2017, through January 1, 2023.

### Viral RNA extraction and quantification
Nasopharyngeal specimens stored in viral transport media were utilized for viral RNA extraction using the QIAamp Viral RNA Minikit (Qiagen, cat. no. 52906) and the QIAcube HT Kit (Qiagen, cat. no. 51331). Viral load quantification and RSV genotype identification were obtained by performing quantitative reverse transcription and PCR (qRT-PCR) using primer and probe set from the World Health Organization Strategy for Global RSV Surveillance Project based on the Influenza Platform[20]. Primers for RSV polymerase [L] (Forward: 5′-AATACAGCCAAATCTAACCAACTTTACA-3′ and Reverse: 5′-GCCAAG-GAAGCATGCAATAAA-3′) used to target both RSV-A and RSV-B, whereas probes (RSV-A: 5′-TGCTATTGTGCACTAAAG-3′ & RSV-B: 5′-CACTATTCCTTACTAAAGATGTC-3′) distinguish between genotype.

### cDNA synthesis and viral genome amplification
cDNA synthesis was performed using SuperScript IV First-Strand Synthesis Kit (Invitrogen, cat. no. 18091050) using RSV-specific primers (Supplementary Table 2)[54]. Due to amplicon dropout in the 2022–2023 RSV season, a Twist Pan-viral panel (Twist Library Preparation EF Kit 2.0 and Twist Comprehensive Viral Research Panel [Twist, SKU no. 103548]) for the enrichment of RSV was used for subsequent degenerate design (Forward: 5′-TATAGGCATGCACCYC-CYTAT-3′ and Reverse: 5′-ACGAGAAAAAAAGTGTYAAAAACT-3′) per manufacturer's instruction. Direct amplification of RSV genome cDNA was performed in separate PCR reactions to generate ~4000 base pair fragments that span the genome. PCR amplification of the generated cDNA was performed using the Phusion Hot Start Flex system (New England Biolabs, cat. no. M0536L). Genomic amplification was confirmed via agarose gel electrophoresis, followed by reaction pooling and reaction cleanup using MAGwise Paramagnetic Beads (seqWell, cat. no. ABIN7271581). After verification of genome amplification, the four amplicon sets were pooled before sequencing library preparation.

### Sequencing library preparation and whole genome sequencing
Two separate approaches to library preparation for sequencing on the Illumina MiSeq platform were used during this study, specifically seqWell's plexWell and expressPlex library preparation. For the seqwell protocol, up to 384 ng of DNA was used for the total Viral genome sequencing reads were utilized to generate consensus sequences using the HAplotype and PHylodynamics pipeline (HAPHPIPE) for viral assembly, population genetics, and phylodynamics (https://github.com/gwcbi/haphpipe)[55]. This pipeline consists of the following: (1) trimming sequencing reads to remove both adapters and low-quality sequences using Trimmomatic v1.0 and (2) three-step assembly refinement. The generated consensus reads were then annotated using Viral Genome ORF Reader (VIGOR) v. 4.0[56].

### Phylogenetic analyses
Generated consensus sequences and ORFs were aligned using MAFFT v7.490, followed by inspection and removal of poorly aligned positions using the Gblocks[57] feature in Seaview v. 5.0.4[58]. IQ-Tree's v. 2.2.0 ModelFinder function[59] was used for the selection of the nucleotide substitution model best fitted for each alignment and subsequent inference of Maximum Likelihood (ML) phylogeny. Assessment of tree topology was done ultrafast bootstrap (UFboot) and Shimodaira–Hasegawa approximate likelihood-ratio test (SH-aLRT) each with 1000 replications. Ancestral sequence reconstruction was performed using TreeTime v. 0.8.5. Likewise, TreeTime was used to estimate time-scaled phylogenies, estimate the evolutionary rate, and calculate the root-to-tip correlation by incorporating sequence sampling dates. Both global (RSV-A = 860, RSV-B = 613) and USA(RSV-A = 340, RSV-B = 113) used in phylogenetic analysis from NCBI Genbank with sample dates up to May 2020 were scanned for high coverage >90%. Due to the low number of publicly available genome sequences after May 2020, all genomes were obtained from the Nextstrain RSV-A and RSV-B databases. ORF analysis was done by extracting cds information using VIGOR v4.1.20200702. which was then used for subsequent phylogenetic and selection analysis.

### Phylodynamic modeling
Global RSV-A and RSV-B genomes with complete date information from NCBI including our genomes were aggregated to perform Bayesian time-scaled phylogenetic analysis after filtering outliers that did not fit the root-to-tip correlation performed with TreeTime (1970 to 2022) and removing alignment positions with poor alignment quality using the default options in Gblocks implemented in Seaview v5.0.4. BEAST 2 v2.7.6[60] was used to estimate each subtypes' rate of evolution and identify the location and time of the MRCA. BEAST 2 priors were introduced with BEAUTI v2.7.6 including a strict molecular clock

model with a lognormal distribution of the evolutionary rate, with a prior for the mean of $4.42 \times 10^{-4}$ substitutions per site per year for RSV-A and $4.22 \times 10^{-4}$ for RSV-B, and a standard deviation of 0.2 for both after optimization with preliminary runs. We assumed a GTR substitution model with invariant sites, as the best-fitted model obtained with ModelFinder, and a Coalescence Bayesian Skyline to model the population size changes through time. Markov chain Monte Carlo (MCMC) runs of at least 100 million states with sampling every 5000 steps were computed. The convergence of MCMC chains was monitored using Tracer v.1.7.2, ensuring that the effective sample size (ESS) values were greater than 200 for each parameter estimated. Phylogeographic patterns were estimated using a discrete-state continuous-time Markov chain to reconstruct the spatial dynamics between geographical locations[61], assuming an asymmetric transition model with separated rate parameters for each possible transition. MCMC was run for over 100 million steps with a burn-in of 20%. Parameters were sampled every 5000 steps and trees were sampled every 10,000 steps. For both RSV-A and RSV-B phylogenies, the maximum clade credibility (MCC) trees were obtained from the tree posterior distribution using TreeAnnotator v2.7.6 after 20% burn-in.

### Structural modeling of the B5.0.5a F protein
A model of GB5.0.5a Fusion pre-fusion conformation protein used to map contemporary mutations was built using SwissModel[62] using the PDB file of RSV strain B18537 Prefusion-stabilized glycoprotein F Variant DS-Cav1 (PDB: 6Q0S.1.a) with as a template. ViralVar[63] was then utilized to visualize the mutation frequencies of isolates sampled after 2020. PyMOL v. 2.4.1 was used to visualize and annotate the following in RSV-B Fusion: (1) antigenic sites (2) predominant mutations appearing after 2020, and (3) mutations at binding sites of monoclonal antibodies nirsevimab, suptavumab, palivizumab, and clesrovimab.

### Statistical analyses
Statistical analyses were performed in R version 4.2.2 and Python version 3.9.5. All multivariable logistic regressions were performed with the glm base function including all indicated variables using $p < 0.05$ to assign statistically significant variables associated with differences between RSV-A and RSV-B cases. The pre-pandemic model was generated by excluding all cases after March 1st, 2020. Package gtsummary v1.7.2 was used to generate descriptive tables, OddsPlottly v1.0.2 to plot odds ratios of the models, and ggplot2 v3.4.4 and ggpubr v0.6.0 were used to generate the statistical figures. Package lifelines v.0.27.8 was used to generate and fit the Kaplan–Meier estimate on the clinical metadata when comparing RSV subtypes in patient length of hospitalization. Hypothesis Testing Using Phylogenies (HyPhy) v.2.4.0 was used to execute both a MEME and a Fast, Unconstrained Bayesian AppRoximation (FUBAR) to determine statistically significant sites undergoing positive selection, as measured by using $p < 0.05$ and posterior probability of positive selection >0.900, respectively.

### Reporting summary
Further information on research design is available in the Nature Portfolio Reporting Summary linked to this article.

## Data availability
Historical national epidemiological data obtained from NREVSS is available at https://data.cdc.gov/Laboratory-Surveillance/Respiratory-Syncytial-Virus-Laboratory-Data-NREVSS/52kb-ccu2/about_data and hospitalization data from RSV-NET is available at https://data.cdc.gov/Public-Health-Surveillance/Weekly-Rates-of-Laboratory-Confirmed-RSV-Hospitali/29hc-w46k/about_data. More recent epidemiological data outside the timeframe of the aggregated NREVSS dataset Is accessible in the CDC NREVSS National Trends (https://www.cdc.gov/surveillance/nrevss/rsv/natl-trend.htm). CDPH epidemiological data were provided by the Vaccine Preventable Disease Surveillance

Program. Accession numbers for the RSV genomes sequenced in this study are listed in Supplementary Table 1. Accession numbers for the publicly available genomes used for phylogenetic analysis are listed in the GitHub repository corresponding to this study (https://github.com/erg6437/RSV-Molecular-Epidemiology/tree/main/Sequence-Info). The prefusion RSV fusion structure (6Q0s) used for Swissmodel modeling can be found on the RCSB Protein Data Bank. Datasets including. individualized clinical data cannot be made publicly available in compliance with IRB protocol. All other datasets are available on the cited RSV Molecular Epidemiology GitHub. However, any other requests for aggregated and de-identified datasets can be made to judd.hultquist@northwestern.edu within 3 years of publication. The amount and format of the requested data that can be shared are dictated by the data-sharing agreements and IRB protocols of each respective institution.

## Code availability
Code for all figures, tree files, BEAST XML files, BEAST log files, and raw data are available at https://github.com/erg6437/RSV-Molecular-Epidemiology [https://doi.org/10.5281/zenodo.10910413][64].

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

## Acknowledgements

The authors are grateful for the assistance from Maryam Shaaban and Dulce Garcia for their assistance with specimen banking. This research was supported in part through the computational resources and staff contributions provided by the Genomics Compute Cluster which is jointly supported by the Feinberg School of Medicine, the Center for Genetic Medicine, and Feinberg's Department of Biochemistry and Molecular Genetics, the Office of the Provost, the Office for Research, and Northwestern Information Technology. The Genomics Compute Cluster is part of Quest, Northwestern University's high-performance computing facility, to advance genomics research. The storage and access to the clinical metadata in the manuscript was supported in part by the Northwestern Medicine Enterprise Data Warehouse, supported by Northwestern University Clinical & Translational Sciences Institute (NUCATS - UL1TR001422). We thank the authors and institutions who deposited RSV sequences via NIH NCBI, and who were instrumental in the larger national and international analyses. We additionally thank the Chicago Department of Public Health Vaccine-Preventable Disease Surveillance unit, particularly Enrique Ramirez, for sharing RSV laboratory data in Chicago. Funding for this work was provided by: the Quantitative Biosciences Institute Coronavirus Research Group (QCRG) Antiviral Drug Discovery (AViDD) center (NIH U19AI171110), the Successful Clinical Response to Pneumonia Therapy (SCRIPT) Center (NIH U19AI135964), and by institutional support for the Center for Pathogen Genomics and Microbial Evolution. E.R.G. was supported by (NIH T32AI007476). The funding sources had no role in the study design, data collection, analysis, interpretation, or writing of the report.

## Author contributions

Conceptualization, E.R.G., L.M.S., H.H.N., M.G.I., R.L.R., E.A.O., J.F.H.; Methodology, E.R.G., L.M.S., R.L.R., E.A.O., J.F.H.; Software, E.R.G., R.L.R., A.A.; Validation, E.R.G, R.L.R., E.A.O., J.F.H.; Formal Analysis, E.R.G., R.L.R., J.F.H.; Investigation, E.R.G., L.M.S., T.J.D., F.M.A.; Resources, L.M.S., M.G.I., E.A.O., J.F.H.; Data Curation, E.R.G, L.M.S., A.P.; Writing – Original Draft, E.R.G., J.F.H.; Writing – Review & Editing, E.R.G., J.F.H.; Visualization, E.R.G., R.L.R., A.A., J.F.H.; Supervision, R.L.R., E.A.O., J.F.H.; Project Administration, L.M.S., R.L.R., E.A.O., J.F.H.; Funding Acquisition, E.A.O., M.G.I., J.F.H.

## Competing interests

M.G.I. declares that research support from GSK was paid to his previous institution, Northwestern University; he received consulting fees from Adagio Therapeutics, ADMA Biologics, Adamis Pharmaceuticals, AlloVir, Atea, Cidara Therapeutics, Genentech/Roche, Janssen, Shionogi, Takeda, Talaris, and Eurofins Viracor; and payment for participating in data safety monitoring boards or advisory boards from Adamis Pharmaceuticals, AlloVir, National Institutes of Health, CSL Behring, Janssen, Merck, Seqirus, Takeda, and Talaris; all of these ended in December 2022; M.G.I. also receives author royalties from UpToDate, which is ongoing. J.F.H. received research support, paid to Northwestern University, from Gilead Sciences and is a paid consultant for Merck. E.R.G has previously been a paid consultant for Merck. All other authors declare no conflicts of interest.
