## [Peer Review File · Nature Communications]

Deviations in RSV epidemiological patterns and population structures in the United States following the COVID-19 pandemicREVIEWER COMMENTS

Reviewer #1 (Remarks to the Author):

In this manuscript, Rioz-Guzman et al. present an extensive analysis of RSV epidemiology and genetic diversity before and during the COVID-19 pandemic. The authors augmented their single-centre retrospective cohort study in Illinois, encompassing clinical, epidemiological, and viral genomes to address specific hypotheses, with data from two national resources, including the National Respiratory and Enterovirus Surveillance and the RSV Hospitalization Surveillance Network across 12 states, offering a comprehensive overview of RSV trends.

A key thesis of the epidemiology is centred around the assertion that changes in RSV testing post-COVID-19 drove increased detections observed in outpatient clinics. Using clinical data, the authors demonstrated a higher risk of intensive hospitalization for adults with RSV-A compared to RSV-B. Leveraging genetic data, they illustrated changes in population structure, disparities between RSV-A and RSV-B, and detailed descriptions of observed mutations.

While the manuscript adeptly addressed various issues and included compelling figures from multiple datasets, there is uncertainty about whether the presented evidence sufficiently substantiates each of the authors' conclusions across the multiple aspects explored in this study. While overall trends are clear, clarifications are needed on the effects of NPI, the scale of the individual analysis (regional, national, global), the relationship between testing and detections, and the acknowledgement of testing variability, which are crucial for enhancing the manuscript's overall quality.

Major comments.

1. The title and abstract do not accurately capture the manuscript's contents. They present findings as universal for RSV, while the study primarily addresses regional patterns under a specific timeline of COVID-19-related NPIs. Detailed clarifications on the scale of the study are necessary.
2. The introduction extensively discusses monoclonal antibodies-based preventative measures, diverting attention from the primary focus on RSV circulation and the impact of NPIs. Including information on RSV resurgence globally following the lifting of control measures would enrich the background on RSV dynamics and their relationship to COVID-19-related NPIs.
3. The first paragraph describing national testing, detections, and positivity lacks clarity, particularly in asserting that increased testing captured an off-season peak. The implications behind changes in testing need clarification. Figure 1 could be enhanced with vertical bars denoting seasonality.
4. The central aim is to understand the impact of NPIs on RSV epidemiology, but the timeline of changes in NPIs at the study site or regionally is not presented.
5. It remains unclear whether the authors addressed the first objective of testing the impact of NPIs, considering extensive changes in testing and sampling.

Minor.

- * The term 'Chicagoland' may be informal.
- 'epidemiological behavior' in line 127 could be simplified to 'epidemiology' for better clarity.

Reviewer #2 (Remarks to the Author):

The authors wrote an interesting article about the RSV epidemiology, clinical severity, and genetic diversity in the years surrounding the COVID-19 pandemic. They found that after 2020 more outpatients with RSV were identified and they related this to increased testing from 2020 onwards since the proportion of RSV positive tests remained the same. They observed an increase in number of RSV positive hospitalized patients after 2020, which was mainly due to an increase in the youngest age group (0-4 years), and was less pronounced in adults. They also compared

disease severity of RSV-A versus B in hospitalized adults and found that RSV-A was associated with more intensive care admissions.

Lastly, they performed whole genome sequencing in a subset of 551 RSV positive patients with available samples of which 218 whole genome sequences were obtained. They found that RSV-B sequences after 2020 formed a monophyletic cluster, which was also seen in other parts of the USA. They also characterized global population dynamics of RSV-B and compared it with RSV-A. They found confirmed strong statistical support for a monophyletic origin of most of the currently circulating RSV-B viruses in the US with a most likely local origin of a common ancestor in 2018. This was not the case for RSV-A.

In addition in RSV-B isolates they looked at the binding sites for monoclonal antibodies and found a high frequency of mutations leading to treatment resistance at the suptavumab binding site and an increase in several mutations at the nirsevumab binding site since 2020 while mutations at the Clesrovimab and Palivizumab binding sites remained low.

The manuscript is well written and contains a lot of information. The data about genomic sequencing is very interesting and might have implications for the future if monoclonal antibodies will be used more widely in infants. The focus of the manuscript is on the adult population, with some comparisons with data from children. It is not clear from the text whether the whole genome sequencing is also partly done in children and if yes, it might be interesting to compare these with adult samples.

Major

Line 221-6 To recover subtyping information, we collected 551 residual diagnostic nasopharyngeal swabs from RSV-positive patients...` and suppl figure 2.

It is not clear if the 551 patients, from whom nasopharyngeal swabs were obtained and whole genome sequencing was initiated, were only adults or also children. When reading the text it seems that all samples were obtained from adults, but if you look in supplemental figure 2, part of the whole genome sequences are from the pediatric population.

Could the authors clarify this and adjust this in the manuscript. I would also recommend to add suppl figure 2 as figure to the manuscript to make it more clear how samples and patients were selected.

Line 363-5 In total, we obtained 218 whole genome sequences (110 subtype A and 108 subtype B) with a minimum of 90% coverage across the genome.

In total 218 samples collecting during 2017-2023 could be sequenced. This means that approximately 35-40 samples per year were sequenced. Could the authors give some more details about the specific season and year and population (adults/children) to see how the sequenced samples were distributed over the years. It would be insightful to see whether samples were collected evenly over the seasons or within a short time interval.

Line 215-16

When controlling for clinical and demographic confounders, RSV-A has historically been associated with increased severity and worse patient outcomes compared to RSV-B.

In literature there is no clear consensus that RSV A is more severe than RSV B in infants, there are also studies which found no difference in severity. Could the authors please elaborate on this. In addition, why did the authors not look in their own population of children/infants whether disease severity differed between RSV A and B and compare this with the adults in the same population?

Line 383-4 Increased transmission and skewed age distributions in the 2021-2023 RSV seasons support this association between increased incidence and waned population immunity in infants.

Please add more information or a reference for the skewed age distribution, I could not find any information about changes in the proportion of infants (0-1 years) in the results.

Minor

Line 97 ...in infants from vaccinated pregnant individuals by 81.8% after the first season of observation

This was efficacy of very severe RSV LRTI after 90 days, not the first season. Could the authors please adjust this.

Line 368-9

...were associated with increased outpatient detections, a factor that might contribute to the increased case counts and altered seasonality observed in recent seasons.

I do not understand how increased testing in outpatients would have led to altered seasonality. Could the authors please explain this?

Figure 1b. The x-axis is showing the years with the incidence of RSV hospitalization. It is not clear during which part of the year the peaks were seen. Could the authors adjust this?

Figure 3.

Figure 3b. One bar part of the histogram is light blue instead of green. Please correct

Figure 3d. The figure seems blurred.

Figure 3f. If numbers are added the total count is 982 instead of 942, which is mentioned in the legend. Please correct.

Table 1.

- From the table and legend it is not clear that only adults were included, which I assumed from the manuscript. Could the authors add this for clarity or adjust if not correct.

- Displaying a mean for age implies that age is distributed normally, but for that a mean age of 62 seems a bit high. Could the authors confirm that age at admission is normally distributed?

Otherwise the authors might consider to add median age.

Reviewer #3 (Remarks to the Author):

The manuscript by Guzman et al. describes the epidemiology of RSV and the potential impact of COVID-19 NPI on RSV genetic diversity in the USA. This manuscript is well written and provides impactful analysis of RSV genomic diversity, particularly as novel RSV vaccines and therapeutics are introduced. Another major strength of the results presented is the analysis of adult infections and emphasis on RSV-B, which are not as well investigated in other studies.

Major comments

Although the authors are clear that one of the limitations of this study is the limited geographic representation of the RSV case data and genomes produced. The RSV genomes produced (n=217, supplementary figure 2) were modest compared the biobank outlined (n=551, ~40% success rate) and the reasons for this low success rate are not clear. Further details of the limitations of the whole genome amplification approach could aid the development of more comprehensive genomic surveillance systems for RSV. Indeed, the original WGS RT primers are not listed in cited reference (40) nor are the primers used to produce the ~4000bp amplicons. For others to replicate this amplification approach the primer sequences would need to be included.

The manuscript calculates if there is an increased risk of severe disease association with RSV-A compared to RSV-B, which has been reported by others, particularly in paediatric cohorts. However, the odds ratio obtained in the study is <1 indicating that RSV-A has lower odds of causing ICU admission than RSV-B. Potentially this is my misinterpretation of the statistical analysis, could the authors please clarify. Also the increased odds of death associated with RSV-A is interesting, which is not significant due to the wide CI. Could the authors discuss the potential

reasons why there are decreased odds of RSV-A causing ICU admission but increased odds of RSV-A associated death.

Minor Comments.

Figure 2a Consider adding titles or a smaller USA map showing Illinois/Chicago for reference to the geographic region sampled in the study.

Figure 3f. Consider revising legend to ease interpretation, assuming yes=ICU admission and no=outpatient or hospitalisation only.

Figure 4. Consider revising figure label "NM" to Illinois/Chicago to ease international interpretation.

Figure 4. Consider using a rectangular tree layout to highlight the topology differences between RSV-A and B discussed.

Figure 5b. Consider annotating the position of the antigenic sites on the F gene cartoon as this would aid comparability between the sequence and structural analysis.

Reviewer #3 (Remarks on code availability):

The github repository contains essential code to replicate many of the analysis conducted in the manuscript. It is clear and easy to navigate.

However it is important that the RSV data is made available, using the ncbi id provided (supplementary table 1) I could not find the sequence data on NCBI. Adding Sup Table 1. to the github repository with SRA/BioProject/Nucleotide Accession details would ease access to the sequencing data produced in the study.

REVIEWER COMMENTS

Reviewer #1 (Remarks to the Author):

In this manuscript, Rios-Guzman et al. present an extensive analysis of RSV epidemiology and genetic diversity before and during the COVID-19 pandemic. The authors augmented their single-centre retrospective cohort study in Illinois, encompassing clinical, epidemiological, and viral genomes to address specific hypotheses, with data from two national resources, including the National Respiratory and Enterovirus Surveillance and the RSV Hospitalization Surveillance Network across 12 states, offering a comprehensive overview of RSV trends.

A key thesis of the epidemiology is centered around the assertion that changes in RSV testing post-COVID-19 drove increased detections observed in outpatient clinics. Using clinical data, the authors demonstrated a higher risk of intensive hospitalization for adults with RSV-A compared to RSV-B. Leveraging genetic data, they illustrated changes in population structure, disparities between RSV-A and RSV-B, and detailed descriptions of observed mutations.

While the manuscript adeptly addressed various issues and included compelling figures from multiple datasets, there is uncertainty about whether the presented evidence sufficiently substantiates each of the authors' conclusions across the multiple aspects explored in this study. While overall trends are clear, clarifications are needed on the effects of NPI, the scale of the individual analysis (regional, national, global), the relationship between testing and detections, and the acknowledgement of testing variability, which are crucial for enhancing the manuscript's overall quality.

Major comments.

1. The title and abstract do not accurately capture the manuscript's contents. They present findings as universal for RSV, while the study primarily addresses regional patterns under a specific timeline of COVID-19-related NPIs. Detailed clarifications on the scale of the study are necessary.

We thank the reviewer for their suggestion given that we draw our data from several different sources. Although the clinical and epidemiological data draw primarily from regional and national sources, the genomic and mutational analyses reflect a combination of regional, national, and global sequences. We have made sure to clarify the spatiotemporal parameters of each analysis in the corresponding figure and figure legend. In addition, we have adjusted the Abstract (Lines 43 – 55) and Title ("Deviations in RSV Epidemiological Patterns and Population Structures in the United States Following the COVID-19 Pandemic") to better reflect the scope.

2. The introduction extensively discusses monoclonal antibodies-based preventative measures, diverting attention from the primary focus on RSV circulation and the impact of NPIs. Including information on RSV resurgence globally following the lifting of control measures would enrich the background on RSV dynamics and their relationship to COVID-19-related NPIs.

We agree with the reviewer that a more extensive explanation of RSV resurgence following the easing of NPIs is needed. We have added a few sentences discussing the global resurgence and country-specific shifts in incidence and seasonality patterns in the Introduction (Lines 98 – 103), Results (Lines 170 – 190), and Discussion (Lines 526 – 540).

3. The first paragraph describing national testing, detections, and positivity lacks clarity, particularly in asserting that increased testing captured an off-season peak. The implications behind changes in testing need clarification. Figure 1 could be enhanced with vertical bars denoting seasonality.

To address these points, we have included more context of standard RSV seasonality and testing patterns before the COVID-19 pandemic (Lines 170 – 174) and clarified the rest of the paragraph. To assist with the visual representation of these trends, we amended Figure 1A per the reviewer's suggestion to depict months on the x-axis with vertical gray bars denoting typical seasons. We discuss the implications of the changes in testing patterns more thoroughly in the Discussion, where we mention outpatient capture, incidental detections, and shifts in societal norms (Lines 544 – 553).

4. The central aim is to understand the impact of NPIs on RSV epidemiology, but the timeline of changes in NPIs at the study site or regionally is not presented.

This is a great suggestion; we have incorporated a rough timeline of NPI implementation in Chicago in revised Figure 2B. These public health mitigation strategies are also now briefly discussed in the Results (Lines 241 – 246) and in the Discussion (Lines 526 – 530).

5. It remains unclear whether the authors addressed the first objective of testing the impact of NPIs, considering extensive changes in testing and sampling.

Our first study objective was to elucidate shifts in RSV epidemiology in Chicago during and after the implementation of COVID-19-related NPIs. While we lack the power to firmly establish causation, we believe that the depth of data in our single-center study illuminates additional underappreciated correlates, especially regarding changes in testing platforms and practices. We believe that by addressing the reviewer's comments above, we have substantially improved the context and clarity of these findings. We have further supplemented these findings by adding more discussion of other studies that explored the impact of NPIs on RSV seasonality (Lines 96-103). We appreciate these suggestions and think these changes have made it a much stronger paper!

Minor Comments.

* The term 'Chicagoland' may be informal.

We have removed the term "Chicagoland" and replaced it with "regional" and "city-wide" throughout the text to be more formal and applicable to a broader audience.

- 'Epidemiological behavior' in line 127 could be simplified to 'epidemiology' for better clarity.

We thank the reviewer for their suggestion and have corrected the phrasing for clarity (Line 162).

Reviewer #2 (Remarks to the Author):

The authors wrote an interesting article about the RSV epidemiology, clinical severity, and genetic diversity in the years surrounding the COVID-19 pandemic. They found that after 2020 more outpatients with RSV were identified and they related this to increased testing from 2020 onwards since the proportion of RSV positive tests remained the same. They observed an increase in the number of RSV positive hospitalized patients after 2020, which was mainly due to an increase in the youngest age group (0-4 years), and was less pronounced in adults. They also compared disease severity of RSV-A versus B in hospitalized adults and found that RSV-A was associated with more intensive care admissions.

Lastly, they performed whole genome sequencing in a subset of 551 RSV positive patients with available samples of which 218 whole genome sequences were obtained. They found that RSV-B sequences after 2020 formed a monophyletic cluster, which was also seen in other parts of the USA. They also characterized global population dynamics of RSV-B and compared it with RSV-A. They found confirmed strong statistical support for a monophyletic origin of most of the currently circulating RSV-B viruses in the US with a most likely local origin of a common ancestor in 2018. This was not the case for RSV-A.

In addition in RSV-B isolates they looked at the binding sites for monoclonal antibodies and found a high frequency of mutations leading to treatment resistance at the suptavumab binding site and an increase in several mutations at the nirsevumab binding site since 2020 while mutations at the Clesrovumab and Palivizumab binding sites remained low.

The manuscript is well written and contains a lot of information. The data about genomic sequencing is very interesting and might have implications for the future if monoclonal antibodies will be used more widely in infants. The focus of the manuscript is on the adult population, with some comparisons with data from children. It is not

clear from the text whether the whole genome sequencing is also partly done in children and if yes, it might be interesting to compare these with adult samples.

Major Comments.

Line 221-6: “To recover subtyping information, we collected 551 residual diagnostic nasopharyngeal swabs from RSV-positive patients...” and suppl figure 2. It is not clear if the 551 patients, from whom nasopharyngeal swabs were obtained and whole genome sequencing was initiated, were only adults or also children. When reading the text it seems that all samples were obtained from adults, but if you look in supplemental figure 2, part of the whole genome sequences are from the pediatric population. Could the authors clarify this and adjust this in the manuscript. I would also recommend to add suppl figure 2 as figure to the manuscript to make it more clear how samples and patients were selected.

We thank the reviewer for this suggestion and apologize for the confusion. We have modified Supplementary Figure 2 to include a bar chart that breaks down our specimen collection by month/year, adult vs. pediatric, and successfully sequenced versus not. We further clarify this in the text where we describe of specimen collection (Lines 303 – 359).

Line 363-5: “In total, we obtained 218 whole genome sequences (110 subtype A and 108 subtype B) with a minimum of 90% coverage across the genome.” In total 218 samples collecting during 2017-2023 could be sequenced. This means that approximately 35-40 samples per year were sequenced. Could the authors give some more details about the specific season year and population (adults/children) to see how the sequenced samples were distributed over the years. It would be insightful to see whether samples were collected evenly over the seasons or within a short time interval.

We thank the reviewer for pointing out the importance of representative sequence distribution over time. Our overall sample collection was roughly proportional to patient encounters (**Figure 2C**) within the Northwestern Medicine system. To make this more clear, we amended **Supplementary Figure 2** to explicitly show the monthly distribution of specimen collection, pediatric/adult isolates, and sequenced genomes throughout our sampling period from 2017 – 2023. We would note that this study was primarily designed to collect residual diagnostic specimens testing positive for RSV from adult populations, but that we did get some pediatric specimens incidentally, especially in later years as routing of diagnostic specimens within the hospital system changed during the COVID-19 pandemic.

Line 215-16: “When controlling for clinical and demographic confounders, RSV-A has historically been associated with increased severity and worse patient outcomes compared to RSV-B.” In the literature there is no clear consensus that RSV A is more severe than RSV B in infants, there are also studies which found no difference in severity. Could the authors please elaborate on this. In addition, why did the authors not look in their population of children/infants whether disease severity differed between RSV A and B and compare this with the adults in the same population?

We apologize if our original phrasing did not adequately highlight the complexity and current lack of consensus regarding the relationship between subtype and severity. We recognize that many factors in the study design, including patient inclusion criteria, severity metrics, and statistical analyses, may drive differences in study conclusions. We have modified the language in this section to reflect this (Line 296 - 301).

In regards to a complementary analysis looking at the relationship between subtype and severity in pediatric populations, we were unfortunately limited by data accessibility. While we were allowed to use incidentally collected pediatric specimens for typing and whole genome sequencing, our IRB did not allow for the collection or use of pediatric clinical metadata. Even if we were to amend our protocol and get IRB approval, we would still be limited as pediatric patients requiring inpatient care are typically routed to Lurie Children’s Hospital, which is an independent institution requiring separate approvals. This is something we would love to do in the future, but unfortunately is not possible at this time.

Line 383-4: “Increased transmission and skewed age distributions in the 2021-2023 RSV seasons support this association between increased incidence and waned population immunity in infants.” Please add more

information or a reference for the skewed age distribution, I could not find any information about changes in the proportion of infants (0-1 years) in the results.

We apologize for the oversight. Unfortunately, we were unable to explore this data in our own cohort as a majority of our patient encounters were adults and the incidental pediatric encounters were largely limited to later seasons (2021 – 2023). To address this, we expanded this part of the discussion to showcase observations of skewed age distributions in other single-center studies (Lines 538 - 541).

Minor Comments.

Line 97: "...in infants from vaccinated pregnant individuals by 81.8% after the first season of observation" This was efficacy of very severe RSV LRTI after 90 days, not the first season. Could the authors please adjust this.

We have corrected the efficacy endpoint of maternal vaccination in the text (Line 122).

Line 368-9: "...were associated with increased outpatient detections, a factor that might contribute to the increased case counts and altered seasonality observed in recent seasons." I do not understand how increased testing in outpatients would have led to altered seasonality. Could the authors please explain this?

We apologize for the confusion. When referring to "altered seasonality", we were referring to both the magnitude and the timing of the RSV seasons after 2020. As testing practices shifted (*i.e.*, increased testing overall, especially off-season), we observed an increase in overall detections and a broadening of seasonal peaks. While we think that the increased testing is likely a factor in the increased detections, especially among outpatients, we don't necessarily think this was a major factor in the off-season peak observed during 2021-2022. We have removed "and altered seasonality" for clarity (Line 516).

Figure 1b. The x-axis is showing the years with the incidence of RSV hospitalization. It is not clear during which part of the year the peaks were seen. Could the authors adjust this?

This is a great suggestion also echoed by Reviewer 1. For clarity, we have revised the x-axes in Figures 1A and 1B to include month and year.

Figure 3b. One bar part of the histogram is light blue instead of green. Please correct.

The original figure panel depicted two overlapping, partially transparent histograms that gave the illusion of a blue bar. This was confusing for a few of our readers, so we eliminated the transparency such that the RSV-A histogram is opaque on top. The trend in each is depicted by the overlaid lines.

Figure 3d. The figure seems blurred.

Figure 3D displays the Kaplan-Meier survival curve with 95% confidence intervals, with the same color scheme as RSV-A (blue) and RSV-B (green). Since the confidence intervals are narrow and the probability of discharge between subtypes is highly comparable, the overlapping curves have a blurry appearance.

Figure 3f. If numbers are added the total count is 982 instead of 942, which is mentioned in the legend. Please correct.

We thank the reviewer for catching this typo in Figure 3! The number of people admitted to the ICU with RSV-B should be 143 as reflected in Table 1. We have corrected the number in the figure to reflect the true count.

Table 1. From the table and legend it is not clear that only adults were included, which I assumed from the manuscript. Could the authors add this for clarity or adjust if not correct.

Of course and we apologize for the confusion. We have clarified in the figure legend title to specify that these patient encounters are adult-only (Line 1253).

Table 1. Displaying a mean for age implies that age is distributed normally, but for that a mean age of 62 seems a bit high. Could the authors confirm that age at admission is normally distributed? Otherwise the authors might consider to add median age.

Figure 3b displays the age distribution of our RSV-A and RSV-B adult inpatient encounters. While the data is close to normally distributed, there is a slight left-handed (younger) tail per the reviewer's point. We have altered Table 1 to reflect the median age as recommended. Regardless, our adult inpatient population does skew older.

Reviewer #3 (Remarks to the Author):

The manuscript by Guzman et al. describes the epidemiology of RSV and the potential impact of COVID-19 NPI on RSV genetic diversity in the USA. This manuscript is well written and provides impactful analysis of RSV genomic diversity, particularly as novel RSV vaccines and therapeutics are introduced. Another major strength of the results presented is the analysis of adult infections and emphasis on RSV-B, which are not as well investigated in other studies.

Major comments.

Although the authors are clear that one of the limitations of this study is the limited geographic representation of the RSV case data and genomes produced. The RSV genomes produced (n=217, supplementary figure 2) were modest compared the biobank outlined (n=551, ~40% success rate) and the reasons for this low success rate are not clear. Further details of the limitations of the whole genome amplification approach could aid the development of more comprehensive genomic surveillance systems for RSV. Indeed, the original WGS RT primers are not listed in cited reference (40) nor are the primers used to produce the ~4000bp amplicons. For others to replicate this amplification approach the primer sequences would need to be included.

We thank the reviewer for raising these concerns and we apologize for not making our sequencing strategy more transparent. We have added a new **Supplementary Table 2** that includes all of the primer sequences used in this study. In total, we recovered whole genome sequences (defined as >90% genome coverage) from 218 isolates, but we attempted to generate amplicons from all 551 specimens. The most common reason that whole genome sequences couldn't be generated was due to the drop out of one or more amplicons. One of the primary drivers of this was low viral load [or high cycle threshold (Ct) value] in the specimen. The median Ct value of all successfully sequenced specimens was just below 24 while the median Ct value of the specimens that failed was around 31. If we look at Ct values by the number of fragments successfully amplified (refer to violin plots below) there is a stepwise decrease in Ct value with each additional fragment amplified. While this is a clear trend, there still were instances where low Ct value specimens failed to amplify and vice versa. We believe this is most likely due to mismatches in one or more primers. While we are still attempting to rescue some of these missing fragments, we were able to get a sufficient representation of sequences from each season in our study for phylogenetic analysis (see new **Supplementary Figure 2**).

The manuscript calculates if there is an increased risk of severe disease association with RSV-A compared to RSV-B, which has been reported by others, particularly in pediatric cohorts. However, the odds ratio obtained in the study is <1 indicating that RSV-A has lower odds of causing ICU admission than RSV-B. Potentially this is my misinterpretation of the statistical analysis, could the authors please clarify. Also the increased odds of death associated with RSV-A is interesting, which is not significant due to the wide CI. Could the authors discuss the potential reasons why there are decreased odds of RSV-A causing ICU admission but increased odds of RSV-A associated death.

The logistic regression model shown in **Figure 3** uses RSV-A as a reference group, with the odds ratio being calculated in relation to RSV-B infections. Therefore, we observe a lower odds of ICU admission when an individual experiences an RSV-B infection. We originally modeled RSV-A versus RSV-B to see if there were differences by subtype for any of the variables, but we understand it is not the most intuitive. For increased clarity, we have now incorporated a new logistic regression analysis in a new **Supplementary Figure 5** that models ICU admission directly. This new model more clearly shows that RSV-B infection is associated with a lower odds of ICU admission. In regard to RSV-associated death, the odds ratio trends in the opposite direction, but it is not significant due to the wide confidence interval. This is mainly driven by the low number of instances in our dataset (21 RSV-A deaths and 33 RSV-B deaths per **Table 1**). Given the low instances and the number of variables in the model, we do not consider this trend meaningful, though we will continue monitoring these differences as our dataset expands over time.

Minor Comments.

Figure 2a. Consider adding titles or a smaller USA map showing Illinois/Chicago for reference to the geographic region sampled in the study.

We have added a small map of the US to **Figure 2A** to show the location of Illinois/Chicago for reference.

Figure 3f. Consider revising legend to ease interpretation, assuming yes=ICU admission and no=outpatient or hospitalisation only.

Thank you for the suggestion; we have changed the legend in **Figure 3F** as recommended.

Figure 4. Consider revising figure label "NM" to Illinois/Chicago to ease international interpretation.

Thank you, we have revised the labels in Figure 4 and the figure legend accordingly.

Figure 4. Consider using a rectangular tree layout to highlight the topology differences between RSV-A and B discussed.

Throughout the manuscript, we use rectangular tree layouts for temporal trees (**Figure 5, Supplementary Figure 6, Supplementary Figure 7**) and circular layouts for diversity trees (**Figure 4, Figure 5, Supplementary Figure 8**). For consistency and clarity, we would like to keep these representations standardized and so have opted to maintain the circular layouts in **Figure 4**.

Figure 5b. Consider annotating the position of the antigenic sites on the F gene cartoon as this would aid comparability between the sequence and structural analysis.

This is a great suggestion. We moved the RSV-B fusion domain schematic in **Figure 5B** to be adjacent to the Fusion structures and highlighted the antigenic sites on the domains respective to the coloring scheme used in **Figure 5C**.

Reviewer #3 (Remarks on code availability): The github repository contains essential code to replicate many of the analysis conducted in the manuscript. It is clear and easy to navigate. However, it is important that the RSV data is made available, using the NCBI id provided (supplementary table 1). I could not find the sequence data on NCBI. Adding Sup Table 1. to the GitHub repository with SRA/BioProject/Nucleotide Accession details would ease access to the sequencing data produced in the study.

We completely agree with the reviewer regarding the importance of sequence availability. We initially embargoed the public release of the sequences pending manuscript acceptance. However, we have since requested their immediate release and they are now available through NCBI (Bioproject PRJNA1050228). We have additionally reviewed the GitHub for clarity and have added Supplementary Table 1 to it as requested. Thank you for your attention to this often-overlooked part of manuscripts!

REVIEWERS' COMMENTS

Reviewer #1 (Remarks to the Author):

The authors have substantially addressed each of my comments. I have no further comments

Reviewer #2 (Remarks to the Author):

The revised manuscript has been improved significantly. The authors have satisfactorily addressed most of my comments. I have one remaining minor comment:

Line 97-100

'However, in 2023 the Food and Drug Administration (FDA) approved two first-in-class RSV vaccines, one that is adjuvanted (Arexvy) and one bivalent, unadjuvanted vaccine (Abrysvo), which reduced the risk of RSV-lower respiratory tract infection (LRTI) in older adults by 94.1% (14) and in infants from vaccinated pregnant individuals by 81.8% (15) after 90 days, respectively.'

It is not clear which vaccine is for which risk group and what the corresponding efficacy is. Arexvy has only been approved for older adults and vaccine efficacy was 94% for severe RSV-related lower respiratory tract disease. Abrysvo has been approved for both older adults (vaccine efficacy of 85.7% against RSV LRTI with 3 or more signs or symptoms, Walsh et al NEJM 2023) and pregnant women (vaccine efficacy of 81.8% against medically attended severe RSV LRTI within 90 day after birth).

Reviewer #3 (Remarks to the Author):

Thank you for comprehensively addressing the reviewers comments provided.

Reviewer #3 (Remarks on code availability):

Now includes details of data availability.

REVIEWERS' COMMENTS

Reviewer #1 (Remarks to the Author):

The authors have substantially addressed each of my comments. I have no further comments

We thank reviewer 1 for their thoughtful feedback in the first round of revisions. We agree that their suggestions greatly enhanced the description of the relationship between nonpharmaceutical interventions and altered RSV seasonality.

Reviewer #2 (Remarks to the Author):

The revised manuscript has been improved significantly. The authors have satisfactorily addressed most of my comments. I have one remaining minor comment:

Line 97-100

'However, in 2023 the Food and Drug Administration (FDA) approved two first-in-class RSV vaccines, one that is adjuvanted (Arexvy) and one bivalent, unadjuvanted vaccine (Abrysvo), which reduced the risk of RSV-lower respiratory tract infection (LRTI) in older adults by 94.1% (14) and in infants from vaccinated pregnant individuals by 81.8% (15) after 90 days, respectively.'

It is not clear which vaccine is for which risk group and what the corresponding efficacy is. Arexvy has only been approved for older adults and vaccine efficacy was 94% for severe RSV-related lower respiratory tract disease. Abrysvo has been approved for both older adults (vaccine efficacy of 85.7% against RSV LRTI with 3 or more signs or symptoms, Walsh et al NEJM 2023) and pregnant women (vaccine efficacy of 81.8% against medically attended severe RSV LRTI within 90 day after birth).

We thank the reviewer for pointing out the need to clarify risk groups and corresponding efficacies for the discussed vaccines. We mistakenly forgot to include that older adults are also eligible for Abrysvo. We have now clarified these points in the text (Lines 94 – 98).

Overall, we thank reviewer #2 for providing suggestions that improved our written clarity on sample inclusion criteria and sequencing efforts.

Reviewer #3 (Remarks to the Author):

Thank you for comprehensively addressing the reviewers comments provided.

Reviewer #3 (Remarks on code availability):

Now includes details of data availability.

We thank the reviewer for evaluating our code availability and for emphasizing the importance of transparency in sequencing strategy. We hope that our comprehensive response to the reviewers is equally reflected in the final manuscript file.